# $\Delta$-STN: Efficient Bilevel Optimization for Neural Networks using Structured Response Jacobians

**Juhan Bae**
University of Toronto
Vector Institute
jbae@cs.toronto.edu

**Roger Grosse**
University of Toronto
Vector Institute
rgrosse@cs.toronto.edu

## Abstract

Hyperparameter optimization of neural networks can be elegantly formulated as a bilevel optimization problem. While research on bilevel optimization of neural networks has been dominated by implicit differentiation and unrolling, hypernetworks such as Self-Tuning Networks (STNs) have recently gained traction due to their ability to amortize the optimization of the inner objective. In this paper, we diagnose several subtle pathologies in the training of STNs. Based on these observations, we propose the $\Delta$-STN, an improved hypernetwork architecture which stabilizes training and optimizes hyperparameters much more efficiently than STNs. The key idea is to focus on accurately approximating the best-response Jacobian rather than the full best-response function; we achieve this by reparameterizing the hypernetwork and linearizing the network around the current parameters. We demonstrate empirically that our $\Delta$-STN can tune regularization hyperparameters (e.g. weight decay, dropout, number of cutout holes) with higher accuracy, faster convergence, and improved stability compared to existing approaches.

## 1 Introduction

Tuning regularization hyperparameters such as weight decay, dropout [55], and data augmentation is indispensable for state-of-the-art performance in a challenging dataset such as ImageNet [8, 52, 7]. An automatic approach to adapting these hyperparameters would improve performance and simplify the engineering process. Although black box methods for tuning hyperparameters such as grid search, random search [2], and Bayesian optimization [53, 54] work well in low-dimensional hyperparameter spaces, they are computationally expensive, require many runs of training, and require that hyperparameter values be fixed throughout training.

Hyperparameter optimization can be elegantly formulated as a bilevel optimization problem [5, 15]. Let $\mathbf{w} \in \mathbb{R}^m$ denote parameters (e.g. weights and biases) and $\boldsymbol{\lambda} \in \mathbb{R}^h$ denote hyperparameters (e.g. weight decay). Let $\mathcal{L}_V$ and $\mathcal{L}_T$ denote validation and training objectives, respectively. We aim to find the optimal hyperparameters $\boldsymbol{\lambda}^*$ that minimize the validation objective at the end of training. Mathematically, the bilevel objective can be formulated as follows[1]:

$$\boldsymbol{\lambda}^* = \underset{\boldsymbol{\lambda} \in \mathbb{R}^h}{\arg \min}\, \mathcal{L}_V(\boldsymbol{\lambda}, \mathbf{w}^*) \text{ subject to } \mathbf{w}^* = \underset{\mathbf{w} \in \mathbb{R}^m}{\arg \min}\, \mathcal{L}_T(\boldsymbol{\lambda}, \mathbf{w}) \qquad (1.1)$$

In machine learning, most work on bilevel optimization has focused on implicit differentiation [28, 48] and unrolling [39, 14]. A more recent approach, which we build on in this work, explicitly approximates the *best-response (rational reaction) function* $\mathbf{r}(\boldsymbol{\lambda}) = \arg \min_{\mathbf{w}} \mathcal{L}_T(\boldsymbol{\lambda}, \mathbf{w})$ with a hypernetwork [49, 19] and jointly optimizes the hyperparameters and the hypernetwork [35]. The

hypernetwork approach is advantageous because training the hypernetwork amortizes the inner-loop optimization work required by both implicit differentiation and unrolling. Since best-response functions are challenging to represent due to their high dimensionality, Self-Tuning Networks (STNs) [38] construct a structured hypernetwork to each layer of the neural network, thereby allowing the efficient and scalable approximation of the best-response function.

In this work, we introduce the $\Delta$-STN, a novel architecture for bilevel optimization that fixes several subtle pathologies in training STNs. We first improve the conditioning of the Gauss-Newton Hessian and fix undesirable bilevel optimization dynamics by reparameterizing the hypernetwork, thereby enhancing the stability and convergence in training. Based on the proposed parameterization, we further introduce a modified training scheme that reduces variance in parameter updates and eliminates any bias induced by perturbing the hypernetwork.

Next, we linearize the best-response hypernetwork to yield an affine approximation of the best-response function. In particular, we linearize the dependency between the network's parameters and predictions so that the training algorithm is encouraged to accurately approximate the Jacobian of the best-response function. Empirically, we evaluate the performance of $\Delta$-STNs on linear models, image classification tasks, and language modelling tasks, and show our method consistently outperform the baselines, achieving better generalization performance in less time.

## 2 Background

### 2.1 Bilevel Optimization with Gradient Descent

A bilevel problem (see Colson et al. [5] for an overview) consists of two sub-problems, where one problem is nested within another. The *outer-level problem (leader)* must be solved subject to the optimal value of the *inner-level problem (follower)*. A general formulation of the bilevel problem is as follows:

$$\min_{\mathbf{x} \in \mathbb{R}^h} f(\mathbf{x}, \mathbf{y}^*) \text{ subject to } \mathbf{y}^* = \arg\min_{\mathbf{y} \in \mathbb{R}^m} g(\mathbf{x}, \mathbf{y}), \tag{2.1}$$

where $f, g \colon \mathbb{R}^h \times \mathbb{R}^m \to \mathbb{R}$ denote outer- and inner-level objectives (e.g. $\mathcal{L}_V$ and $\mathcal{L}_T$), and $\mathbf{x} \in \mathbb{R}^h$ and $\mathbf{y} \in \mathbb{R}^m$ denote outer- and inner-level variables (e.g. $\boldsymbol{\lambda}$ and $\mathbf{w}$). Many problems can be cast as bilevel objectives in machine learning, including hyperparameter optimization, generative adversarial networks (GANs) [18, 24], meta-learning [15], and neural architecture search [63, 34, 6].

A naïve application of simultaneous gradient descent on training and validation objectives will fail due to the hierarchy induced by the bilevel structure [13, 59]. A more principled approach in solving the bilevel problem is to incorporate the best-response function. Substituting the best-response function in the validation objective converts the bilevel problem to a single-level problem:

$$\min_{\boldsymbol{\lambda} \in \mathbb{R}^h} \mathcal{L}_V(\boldsymbol{\lambda}, \mathbf{r}(\boldsymbol{\lambda})) \tag{2.2}$$

We refer readers to Fiez et al. [13] on a more detailed analysis of the best-response (rational reaction) function. The Jacobian of the best-response function is as follows:

$$\frac{\partial \mathbf{r}}{\partial \boldsymbol{\lambda}}(\boldsymbol{\lambda}) = -\left(\frac{\partial^2 \mathcal{L}_T}{\partial \mathbf{w}^2}(\boldsymbol{\lambda}, \mathbf{r}(\boldsymbol{\lambda}))\right)^{-1} \frac{\partial^2 \mathcal{L}_T}{\partial \mathbf{w} \partial \boldsymbol{\lambda}}(\boldsymbol{\lambda}, \mathbf{r}(\boldsymbol{\lambda})) \tag{2.3}$$

The gradient of the validation objective is composed of *direct* and *response gradients*. While the direct gradient captures the direct dependence on the hyperparameters in the validation objective, the response gradient captures how the optimal parameter responds to the change in the hyperparameters:

$$\frac{\mathrm{d}\mathcal{L}_V}{\mathrm{d}\boldsymbol{\lambda}}(\boldsymbol{\lambda}, \mathbf{r}(\boldsymbol{\lambda})) = \underbrace{\frac{\partial \mathcal{L}_V}{\partial \boldsymbol{\lambda}}(\boldsymbol{\lambda}, \mathbf{w})}_{\text{Direct Gradient}} + \underbrace{\left(\frac{\partial \mathbf{r}}{\partial \boldsymbol{\lambda}}(\boldsymbol{\lambda})\right)^{\top} \frac{\partial \mathcal{L}_V}{\partial \mathbf{w}}(\boldsymbol{\lambda}, \mathbf{r}(\boldsymbol{\lambda}))}_{\text{Response Gradient}} \tag{2.4}$$

For most hyperparameter optimization problems, the regularization penalty is not imposed in the validation objective, so the direct gradient is $\mathbf{0}$. Therefore, either computing or approximating the best-response Jacobian (**–**) is essential for computing the gradient of the outer objective.

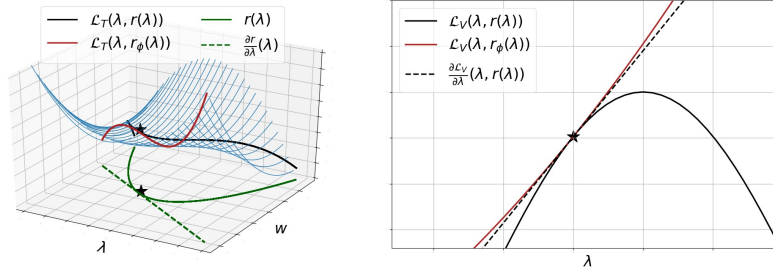

**Figure 1: (Left)** The loss surface of the training objective. The best-response hypernetwork $\mathbf{r}_\phi$ locally approximates the best-response function at the current configuration ($\star$). **(Right)** The loss surface of the validation objective. The local best-response hypernetwork embedded in the validation objective allows to capture the response gradient in Eqn. 2.4.

## 2.2 Best-Response Hypernetwork

Lorraine and Duvenaud [35] locally model the best-response function with a hypernetwork. Let $\phi \in \mathbb{R}^p$ be the best-response (hypernetwork) parameters and $p(\boldsymbol{\epsilon}|\boldsymbol{\sigma})$ be a zero-mean diagonal Gaussian distribution with a fixed perturbation scale $\boldsymbol{\sigma} \in \mathbb{R}_+^h$. The objective for the hypernetwork is as follows:

$$\min_{\boldsymbol{\phi} \in \mathbb{R}^p} \mathbb{E}_{\boldsymbol{\epsilon} \sim p(\boldsymbol{\epsilon}|\boldsymbol{\sigma})} \left[ \mathcal{L}_T(\boldsymbol{\lambda} + \boldsymbol{\epsilon}, \mathbf{r}_\phi(\boldsymbol{\lambda} + \boldsymbol{\epsilon})) \right], \tag{2.5}$$

where $\mathbf{r}_\phi$ is the best-response hypernetwork. Intuitively, the hyperparameters are perturbed so that the hypernetwork can locally learn the best-response curve, as shown in figure 1. Self-Tuning Networks (STNs) [38] extend this approach by constructing a structured hypernetwork for each layer in neural networks in order to scale well with the number of hyperparameters. They parameterize the best-response function as:

$$\mathbf{r}_\phi(\boldsymbol{\lambda}) = \boldsymbol{\Phi}\boldsymbol{\lambda} + \boldsymbol{\phi}_0, \tag{2.6}$$

where $\boldsymbol{\phi}_0 \in \mathbb{R}^m$ and $\boldsymbol{\Phi} \in \mathbb{R}^{m \times h}$, and further impose a structure on weights of the neural network, as detailed in section 3.4. The STN performs alternating gradient descents on hypernetwork, hyperparameters and perturbation scale, where the validation objective is formulated as:

$$\min_{\boldsymbol{\lambda} \in \mathbb{R}^h, \boldsymbol{\sigma} \in \mathbb{R}_+^h} \mathbb{E}_{\boldsymbol{\epsilon} \sim p(\boldsymbol{\epsilon}|\boldsymbol{\sigma})} \left[ \mathcal{L}_V(\boldsymbol{\lambda} + \boldsymbol{\epsilon}, \mathbf{r}_\phi(\boldsymbol{\lambda} + \boldsymbol{\epsilon})) \right] - \tau \mathbb{H}\left[ p(\boldsymbol{\epsilon}|\boldsymbol{\sigma}) \right] \tag{2.7}$$

Here, $\mathbb{H}[\cdot]$ is an entropy term weighted by $\tau \in \mathbb{R}_+$ to prevent the perturbation scale from getting too small. Note that the above objective is similar to that of variational inference, where the first term is analogous to the negative log-likelihood. When $\tau$ ranges from 0 to 1, the objective interpolates between variational inference and variational optimization [56]. The full algorithm for the STN is described in Alg. 2 (appendix B).

## 3 Method

We now describe our main contributions: a centered parameterization of the hypernetwork, a modified training objective, and a linearization of the best-response hypernetwork. For simplicity, we first present these contributions in the context of a full linear hypernetwork (which is generally impractical to represent) and then explain how our contributions can be combined with the compact hypernetwork structure used in STNs.

## 3.1 Centered Parameterization of the Best-Response Hypernetwork

Our first contribution is to use a centered parameterization of the hypernetwork. In particular, observe that if the hyperparameters are transformed as $\widetilde{\boldsymbol{\lambda}} = \boldsymbol{\lambda} + \mathbf{a}$, for any vector $\mathbf{a} \in \mathbb{R}^h$, then the hypernetwork parameters can be transformed as $\widetilde{\boldsymbol{\phi}}_0 = \boldsymbol{\Phi}\mathbf{a} + \boldsymbol{\phi}_0$ to represent the same mapping in Eqn. 2.6. Therefore, we can reparameterize the model by adding an offset to the hyperparameters if it is advantageous for optimization. As it is widely considered beneficial to center inputs and activations at 0 [30, 22, 44], we propose to adopt the following centered parameterization of the hypernetwork:

$$\mathbf{r}_{\boldsymbol{\theta}}(\boldsymbol{\lambda}, \boldsymbol{\lambda}_0) = \boldsymbol{\Theta}(\boldsymbol{\lambda} - \boldsymbol{\lambda}_0) + \mathbf{w}_0, \tag{3.1}$$

where $\mathbf{w}_0 \in \mathbb{R}^m$ and $\boldsymbol{\Theta} \in \mathbb{R}^{m \times h}$. Intuitively, if $\boldsymbol{\lambda}_0$ is regarded as the "current hyperparameters", then $\mathbf{w}_0$ can be seen as the "current weights", and $\boldsymbol{\Theta}$ determines how the weights are adjusted in response to a perturbation to $\boldsymbol{\lambda}$. We provide two justifications why the centered parameterization is advantageous; the first justification relates to the conditioning of the single-level optimization problem for $\phi$, while the second involves a phenomenon particular to bilevel optimization.

For the first justification, consider optimizing the hypernetwork parameters $\phi$ for a fixed $\bar{\boldsymbol{\lambda}} \in \mathbb{R}^{h+1}$, where $\bar{\boldsymbol{\lambda}}$ is defined as a vector formed by appending additional homogeneous coordinate with value 1 to incorporate the offset term. The speed of convergence of gradient descent is closely related to the condition number of the Hessian $\mathbf{H}_\phi = \nabla^2_\phi \mathcal{L}_T$ [47]. For neural networks, it is common to approximate $\mathbf{H}_\phi$ with the Gauss-Newton Hessian [30, 41], which linearizes the network around $\mathbf{w}$:

$$\mathbf{H_w} \approx \mathbf{G_w} \triangleq \mathbb{E}\left[\mathbf{J_{yw}^\top H_y J_{yw}}\right], \tag{3.2}$$

where $\mathbf{J_{yw}} = \partial \mathbf{y}/\partial \mathbf{w}$ is the weight-output Jacobian, $\mathbf{H_y} = \nabla^2_\mathbf{y} \mathcal{L}_T$ is the Hessian of the loss with respect to the network outputs $\mathbf{y}$, and the expectation is with respect to the training distribution.

**Observation 1.** *The Gauss-Newton Hessian with respect to the hypernetwork is given by:*

$$\mathbf{G}_\phi = \mathbb{E}\left[\hat{\boldsymbol{\lambda}}\hat{\boldsymbol{\lambda}}^\top \otimes \mathbf{J_{yw}^\top H_y J_{yw}}\right], \tag{3.3}$$

*where $\hat{\boldsymbol{\lambda}} = \bar{\boldsymbol{\lambda}} + \bar{\boldsymbol{\epsilon}}$ are the sampled hyperparameters and $\bar{\boldsymbol{\epsilon}}$ is the perturbation vector appended with additional homogeneous coordinate with value 0.*

See appendix C.1 for the derivation. Heuristically, we can approximate $\mathbf{G}_\phi$ by pushing the expectation inside the Kronecker product, a trick that underlies the K-FAC optimizer [42, 62]:

$$\mathbf{G}_\phi \approx \mathbb{E}[\hat{\boldsymbol{\lambda}}\hat{\boldsymbol{\lambda}}^\top] \otimes \mathbb{E}\left[\mathbf{J_{yw}^\top H_y J_{yw}}\right] = \mathbb{E}[\hat{\boldsymbol{\lambda}}\hat{\boldsymbol{\lambda}}^\top] \otimes \mathbf{G_w}, \tag{3.4}$$

where $\mathbf{G_w}$ is the Gauss-Newton Hessian for the network itself. We now note the following well-known fact about the condition number of the Kronecker product (see appendix C.2 for the proof):

**Observation 2.** *Let $\kappa(\mathbf{A})$ denote the condition number of a square positive definite matrix $\mathbf{A}$. Given square positive definite matrices $\mathbf{A}$ and $\mathbf{B}$, the condition number of $\mathbf{A} \otimes \mathbf{B}$ is given by $\kappa(\mathbf{A} \otimes \mathbf{B}) = \kappa(\mathbf{A})\kappa(\mathbf{B})$.*

Hence, applying Observation 2 to Eqn. 3.4, we would like to make the factor $\mathbb{E}[\hat{\boldsymbol{\lambda}}\hat{\boldsymbol{\lambda}}^\top]$ well-conditioned in order to make the overall optimization problem well-conditioned. We apply the well-known decomposition of the second moments:

$$\mathbb{E}[\hat{\boldsymbol{\lambda}}\hat{\boldsymbol{\lambda}}^\top] = \text{Cov}(\hat{\boldsymbol{\lambda}}) + \mathbb{E}[\hat{\boldsymbol{\lambda}}]\mathbb{E}[\hat{\boldsymbol{\lambda}}]^\top = \text{Cov}(\boldsymbol{\epsilon}) + \bar{\boldsymbol{\lambda}}\bar{\boldsymbol{\lambda}}^\top \tag{3.5}$$

In the context of our method, the first term is the diagonal matrix $\text{diag}(\boldsymbol{\sigma}^2)$, whose entries are typically small. The second term is a rank-1 matrix whose nonzero eigenvalue is $\|\boldsymbol{\lambda}\|^2$. If $\boldsymbol{\lambda}$ is high-dimensional or far from $\mathbf{0}$, the latter will dominate, and the problem will be ill-conditioned due to the large outlier eigenvalue. Therefore, to keep the problem well-conditioned, we would like to ensure that $\|\boldsymbol{\lambda}\|^2$ is small and one way to do this is to use a centered parameterization that sends $\boldsymbol{\lambda} \mapsto \mathbf{0}$, such as Eqn. 3.1. This suggests that centering should improve the conditioning of the inner objective.

The above analysis justifies why we would expect centering to speed up training of the hypernetwork, taken as a single-level optimization problem. However, there is also a second undesirable effect of uncentered representations involving *interactions* between the inner and outer optimizations. The result of a single batch gradient descent update to $\boldsymbol{\Phi}$ is as follows:

$$\boldsymbol{\Phi}^{(1)} = \boldsymbol{\Phi}^{(0)} - \alpha \nabla_{\boldsymbol{\Phi}}(\mathcal{L}_T(\boldsymbol{\lambda}^{(0)}, \mathbf{r}_{\phi^{(0)}}(\boldsymbol{\lambda}^{(0)}))) = \boldsymbol{\Phi}^{(0)} - \alpha(\nabla_\mathbf{w}\mathcal{L}_T(\boldsymbol{\lambda}^{(0)}, \mathbf{w}^{(0)}))(\boldsymbol{\lambda}^{(0)})^\top, \tag{3.6}$$

where $\boldsymbol{\lambda}^{(0)}$ denotes the hyperparameters sampled in the first iteration and $\alpha$ denotes the learning rate of the inner objective. This results in a failure of credit assignment: it induces the (likely mistaken) belief that adjusting $\boldsymbol{\lambda}$ in the direction of $\boldsymbol{\lambda}$ will move the optimal weights in the direction of $-\nabla_\mathbf{w}\mathcal{L}_T$. Plugging in Eqn. 3.6, this leads to the following hyperparameter gradient $\nabla_{\boldsymbol{\lambda}}(\mathcal{L}_V)$:

$$\nabla_{\boldsymbol{\lambda}}(\mathcal{L}_V(\boldsymbol{\lambda}, \mathbf{r}_{\phi^{(1)}}(\boldsymbol{\lambda}))) = \nabla_{\boldsymbol{\lambda}}\mathcal{L}_V(\boldsymbol{\lambda}, \mathbf{w}^{(1)}) + (\boldsymbol{\Phi}^{(1)})^\top \nabla_\mathbf{w}\mathcal{L}_V(\boldsymbol{\lambda}, \mathbf{w}^{(1)}) \tag{3.7}$$

$$= \nabla_{\boldsymbol{\lambda}}\mathcal{L}_V(\boldsymbol{\lambda}, \mathbf{w}^{(1)}) + (\boldsymbol{\Phi}^{(0)})^\top \nabla_\mathbf{w}\mathcal{L}_V(\boldsymbol{\lambda}, \mathbf{w}^{(1)})$$

$$- \alpha(\nabla_\mathbf{w}\mathcal{L}_T(\boldsymbol{\lambda}^{(0)}, \mathbf{w}^{(0)}))^\top(\nabla_\mathbf{w}\mathcal{L}_V(\boldsymbol{\lambda}, \mathbf{w}^{(1)}))\boldsymbol{\lambda}^{(0)} \tag{3.8}$$

In Eqn. 3.8, the coefficient in front of $\boldsymbol{\lambda}^{(0)}$ in the last term is the inner product between the training and validation gradients. Early in training, we would expect the training and validation gradients to be well-aligned, so this inner product would be positive. Hence, $\boldsymbol{\lambda}$ will tend to move in the direction of $\boldsymbol{\lambda}$, i.e. away from $\mathbf{0}$, simply due to the bilevel optimization dynamics. We found this to be a very strong effect in some cases, and it resulted in pathological choices of hyperparameters early in training. Using the centered parameterization appeared to fix the problem, as we show in section 5.1 and appendix H.1.

## 3.2 Modified Update Rule for Centered Parameterization

We observed in section 3.1 that in the centered parameterization, $\mathbf{w}_0$ can be seen as the current weights, while $\boldsymbol{\Theta}$ is the Jacobian of the approximate best response function $\mathbf{r}_{\boldsymbol{\theta}}$. This suggests a modification to the original STN training procedure. While in the original STN, the full hypernetwork was trained using sampled hyperparameters, we claim that $\mathbf{w}_0$ can instead be trained using gradient descent on the regularized training loss, just like the weights of an ordinary neural network. In this sense, we separate the training objectives in the following manner:

$$\mathbf{w}_0^* = \arg\min_{\mathbf{w}_0 \in \mathbb{R}^m} \mathcal{L}_T(\boldsymbol{\lambda}, \mathbf{w}_0) \tag{3.9}$$

$$\boldsymbol{\Theta}^* = \arg\min_{\boldsymbol{\Theta} \in \mathbb{R}^{m \times h}} \mathbb{E}_{\boldsymbol{\epsilon} \sim p(\boldsymbol{\epsilon}|\boldsymbol{\sigma})} \left[ \mathcal{L}_T(\boldsymbol{\lambda} + \boldsymbol{\epsilon}, \mathbf{r}_{\boldsymbol{\theta}}(\boldsymbol{\lambda} + \boldsymbol{\epsilon}, \boldsymbol{\lambda})) \right] \tag{3.10}$$

The exclusion of perturbation in Eqn. 3.9 reduces the variance for the updates on $\mathbf{w}_0$, yielding faster convergence. Moreover, it can eliminate any bias to the optimal $\mathbf{w}_0^*$ induced by the perturbation. In appendix G, we show that, even for linear regression with $L_2$ regularization, the optimal weight $\mathbf{w}_0^*$ does not match the correct solution under the STN's objective, whereas the modified objective recovers the correct solution. The following theorem shows that, for a general quadratic inner-level objective, the proposed parameterization converges to the best-response Jacobian.

**Theorem 3.** *Suppose $\mathcal{L}_T$ is quadratic with $\partial^2 \mathcal{L}_T / \partial \mathbf{w}^2 \succ 0$ and let $p(\boldsymbol{\epsilon}|\boldsymbol{\sigma})$ be a diagonal Gaussian distribution with mean $\mathbf{0}$ and variance $\sigma^2 \mathbf{I}$. Fixing $\boldsymbol{\lambda}_0 \in \mathbb{R}^h$ and $\mathbf{w}_0 \in \mathbb{R}^m$, the solution to the objective in Eqn. 3.10 is the best-response Jacobian.*

See appendix C.3 for the proof.

## 3.3 Direct Approximation of the Best-Response Jacobian using a Linearized Network

The STN aims to learn a linear hypernetwork that approximates the best-response function in a region around $\boldsymbol{\lambda}$. However, if the perturbation $\boldsymbol{\epsilon}$ is large, it may be difficult to approximate the best-response function as linear within this region. Part of the problem is that the function represented by the network behaves nonlinearly with respect to $\mathbf{w}$, such that the linear adjustment represented by $\boldsymbol{\Phi}\boldsymbol{\lambda}$ (in Eqn. 2.6) may have a highly nonlinear effect on the network's predictions. We claim it is in fact unnecessary to account for the nonlinear effect of large changes to $\mathbf{w}$, as the hypernetwork is only used to estimate the best-response Jacobian at $\boldsymbol{\lambda}_0$, and the Jacobian depends only on the effect of infinitesimal changes to $\mathbf{w}$.

To remove the nonlinear dependence of the predictions on $\mathbf{w}$, we linearize the network around the current weights $\mathbf{r}(\boldsymbol{\lambda}_0) = \mathbf{w}_0$. The first-order Taylor approximation to the network computations is given by:

$$\mathbf{y} = f(\mathbf{x}, \mathbf{w}, \boldsymbol{\lambda}, \xi) \approx f(\mathbf{x}, \mathbf{w}_0, \boldsymbol{\lambda}, \xi) + \mathbf{J}_{\mathbf{yw}}(\mathbf{w} - \mathbf{w}_0), \tag{3.11}$$

where $f(\mathbf{x}, \mathbf{w}, \boldsymbol{\lambda}, \xi)$ denotes evaluating a network with weights $\mathbf{w}$ and hyperparameters $\boldsymbol{\lambda}$ on input $\mathbf{x}$. Here, $\xi$ denotes a source of randomness (e.g. dropout mask) and $\mathbf{J}_{\mathbf{yw}} = \partial \mathbf{y} / \partial \mathbf{w}$ is the weight-output Jacobian. This relationship can also be written with the shorthand notation $\Delta \mathbf{y} \approx \mathbf{J}_{\mathbf{yw}} \Delta \mathbf{w}$, where $\Delta$ denotes a small perturbation. Therefore, we refer to it as the $\Delta$-approximation, and the corresponding bilevel optimization method as the $\Delta$-STN. In the context of our method, given the hyperparameters $\boldsymbol{\lambda}_0 \in \mathbb{R}^h$ and the perturbation $\boldsymbol{\epsilon} \in \mathbb{R}^h$, we structure the prediction as follows:

$$\mathbf{y}' = f(\mathbf{x}, \mathbf{r}_{\boldsymbol{\theta}}(\boldsymbol{\lambda}_0 + \boldsymbol{\epsilon}, \boldsymbol{\lambda}_0), \boldsymbol{\lambda}_0 + \boldsymbol{\epsilon}, \xi) \approx f(\mathbf{x}, \mathbf{r}_{\boldsymbol{\theta}}(\boldsymbol{\lambda}_0, \boldsymbol{\lambda}_0), \boldsymbol{\lambda}_0 + \boldsymbol{\epsilon}, \xi) + \mathbf{J}_{\mathbf{yw}} \Delta \mathbf{w} \tag{3.12}$$

$$= f(\mathbf{x}, \mathbf{w}_0, \boldsymbol{\lambda}_0 + \boldsymbol{\epsilon}, \xi) + \mathbf{J}_{\mathbf{yw}} \boldsymbol{\Theta} \boldsymbol{\epsilon} \tag{3.13}$$

In figure 2, we show the approximated best-response Jacobian obtained by STN and $\Delta$-STN, and compare them with the true best-response Jacobian. The contours show the loss surface of the training objective at some $\lambda$ and we projected best-response Jacobian to $(\lambda, w_1)$ and $(\lambda, w_2)$ planes for a better comparison. Because of the nonlinearity around $\lambda_0$ ($\star$), the STN tries to fit a more horizontal best-response function to minimize the error given by the perturbation, degrading the accuracy of the Jacobian approximation. On the other hand, the $\Delta$-STN linearizes the best-response function at $\lambda_0$, focusing on accurately capturing the linear effects. In appendix D, we show that, with some approximations, the linearization of the perturbed inner objective yields a correct approximation of the best-response Jacobian.

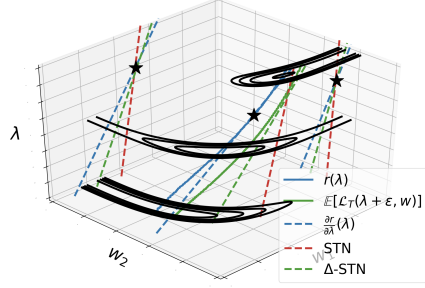

**Figure 2:** A comparison of approximated best-response Jacobians obtained by STN and $\Delta$-STN at $\lambda_0$ ($\star$). The $\Delta$-STN approximates the best-response Jacobian more accurately by linearizing the best-response hypernetwork.

Evaluating $\mathbf{J_{yw}}\Delta\mathbf{w}$ in Eqn. 3.11 corresponds to evaluating the directional derivative of $f$ in the direction $\Delta\mathbf{w}$ and can be efficiently computed using forward mode automatic differentiation (Jacobian-vector products), a core feature in frameworks such as JAX [4].

## 3.4 Structured Hypernetwork Representation

So far, the discussion has assumed a general linear hypernetwork for simplicity. However, a full linear hypernetwork would have dimension $h \times m$, which is impractical to represent if $\boldsymbol{\lambda}$ is high-dimensional. Instead, we adopt an efficient parameterization analogous to that of the original STN. Considering the $i$-th layer of the neural network, whose weights and bias are $\mathbf{W}^{(i)} \in \mathbb{R}^{m_i \times m_{i+1}}$ and $\mathbf{b}^{(i)} \in \mathbb{R}^{m_{i+1}}$, we propose to structure the layer-wise best-response hypernetwork as follows[2]:

$$\mathbf{W}_{\boldsymbol{\theta}}^{(i)}(\boldsymbol{\lambda}, \boldsymbol{\lambda}_0) = \mathbf{W}_{\text{general}}^{(i)} + \left(\mathbf{U}^{(i)}(\boldsymbol{\lambda} - \boldsymbol{\lambda}_0)\right) \odot_{\text{row}} \mathbf{W}_{\text{response}}^{(i)}$$
$$\mathbf{b}_{\boldsymbol{\theta}}^{(i)}(\boldsymbol{\lambda}, \boldsymbol{\lambda}_0) = \mathbf{b}_{\text{general}}^{(i)} + \left(\mathbf{V}^{(i)}(\boldsymbol{\lambda} - \boldsymbol{\lambda}_0)\right) \odot \mathbf{b}_{\text{response}}^{(i)},$$

$$(3.14)$$

where $\mathbf{U}^{(i)}, \mathbf{V}^{(i)} \in \mathbb{R}^{m_{i+1} \times h}$, and $\odot_{\text{row}}$ and $\odot$ denote row-wise and element-wise multiplications. Observe that these formulas are linear in $\boldsymbol{\lambda}$, so they can be seen as a special case of Eqn. 3.1, except that structure is imposed on $\boldsymbol{\Theta}$. Observe also that this architecture is memory efficient and tractable to compute, and allows parallelism: it requires $m_{i+1}(2m_i + h)$ and $m_{i+1}(2 + h)$ parameters to represent the weights and bias, respectively, with 2 additional matrix multiplications and element-wise multiplications in the forward pass. Both quantities are significantly smaller than the full best-response Jacobian, so the $\Delta$-STN incurs limited memory or computational overhead compared with simply training a neural network.

## 3.5 Training Algorithm

---

**Algorithm 1:** Training Algorithm for $\Delta$-STNs

---

**Initialize:** hypernetwork $\boldsymbol{\theta} = \{\mathbf{w}_0, \boldsymbol{\Theta}\}$; hyperparameters $\boldsymbol{\lambda}$; learning rates $\{\alpha_i\}_{i=1}^3$; training and validation update intervals $T_{\text{train}}, T_{\text{valid}}$, and entropy penalty $\tau$.

**while** *not converged* **do**

    **for** $t = 1, ..., T_{train}$ **do**

        $\boldsymbol{\epsilon} \sim p(\boldsymbol{\epsilon}|\boldsymbol{\sigma})$

        $\mathbf{w}_0 \leftarrow \mathbf{w}_0 - \alpha_1 \nabla_{\mathbf{w}_0}(\mathcal{L}_T(\boldsymbol{\lambda}, \mathbf{r}_{\boldsymbol{\theta}}(\boldsymbol{\lambda}, \boldsymbol{\lambda})))$

        $\boldsymbol{\Theta} \leftarrow \boldsymbol{\Theta} - \alpha_1 \nabla_{\boldsymbol{\Theta}}(\mathcal{L}_T(\boldsymbol{\lambda} + \boldsymbol{\epsilon}, \mathbf{r}_{\boldsymbol{\theta}}(\boldsymbol{\lambda} + \boldsymbol{\epsilon}, \boldsymbol{\lambda})))$ ▷ Linearization with forward-mode autodiff

    **for** $t = 1, ..., T_{valid}$ **do**

        $\boldsymbol{\epsilon} \sim p(\boldsymbol{\epsilon}|\boldsymbol{\sigma})$

        $\boldsymbol{\lambda} \leftarrow \boldsymbol{\lambda} - \alpha_2 \nabla_{\boldsymbol{\lambda}}(\mathcal{L}_V(\boldsymbol{\lambda} + \boldsymbol{\epsilon}, \mathbf{r}_{\boldsymbol{\theta}}(\boldsymbol{\lambda} + \boldsymbol{\epsilon}, \boldsymbol{\lambda}_0)))|_{\boldsymbol{\lambda}=\boldsymbol{\lambda}_0}$

        $\boldsymbol{\sigma} \leftarrow \boldsymbol{\sigma} - \alpha_3 \nabla_{\boldsymbol{\sigma}}(\mathcal{L}_V(\boldsymbol{\lambda} + \boldsymbol{\epsilon}, \mathbf{r}_{\boldsymbol{\theta}}(\boldsymbol{\lambda} + \boldsymbol{\epsilon}, \boldsymbol{\lambda})) - \tau \mathbb{H}[p(\boldsymbol{\epsilon}|\boldsymbol{\sigma})])$

---

The full algorithm for $\Delta$-STNs is given in Alg. 1. In comparison to STNs (Alg. 2 in appendix B), we use a modified training objective and a centered hypernetwork parameterization with linearization.

The hyperparameters and perturbation scale are optimized using the same objectives as in STNs (Eqn. 2.7). Since hyperparameters are often constrained (e.g. dropout rate is in between 0 and 1), we apply a fixed non-linear transformation to the hyperparameters and optimize the hyperparameters on an unconstrained domain, as detailed in appendix F.

## 4   Related Work

Automatic procedures for finding an effective set of hyperparameters have been a prominent subject in the literature (see Feurer and Hutter [12] for an overview). Early works have focused on model-free approaches such as grid search and random search [2]. Hyperband [31] and Successive Halving Algorithm (SHA) [32] extend on random search by using multi-armed bandit techniques [23] to terminate poor-performing hyperparameter configurations early. These model-free approaches are straightforward to parallelize and work well in practice. However, they rely on random procedures, not exploiting the structure of the problem.

Bayesian Optimization (BO) provides a more principled tool to optimize the hyperparameters. Given the hyperparameters $\boldsymbol{\lambda}$ and the observations $\mathcal{O} = \{(\boldsymbol{\lambda}_i, s_i)\}_{i=1}^{n}$, where $s$ is a surrogate loss, BO models a conditional probability $p(s|\boldsymbol{\lambda}, \mathcal{O})$ [21, 3, 53, 54]. The observations are constructed in an iterative manner, where the next set of hyperparameters to train the model is the one that maximizes an acquisition function $C(\boldsymbol{\lambda}; p(s|\boldsymbol{\lambda}, \mathcal{O}))$, which trades off exploitation and exploration. The training through convergence may be avoided under some assumptions on the learning curve behavior [57, 25]. Nevertheless, BO requires building inductive bias into the conditional probability, is sensitive to the parameters of the surrogate model, and most importantly, does not scale well with the number of hyperparameters.

In comparison to black-box optimization, the use of gradient information can provide a drastic improvement in convergence [46]. There are two major approaches to gradient-based hyperparameter optimization. The first method uses the implicit function theorem to obtain the best-response Jacobian $\partial \mathbf{r}/\partial \boldsymbol{\lambda}$ [28, 1, 48, 36], which requires approximating the Hessian (or Gauss-Newton) inverse. The second approach approximates the best-response function $\mathbf{r}$ by casting the inner objective as a dynamical system [10, 39, 37, 15, 50] and applying automatic differentiation to compute the best-response Jacobian. Both approaches are computationally expensive: implicit differentiation requires approximating the inverse Hessian and unrolled differentiation needs to backpropagate through the whole gradient steps.

In contrast to implicit differentiation and unrolling, the hypernetwork approach [35] such as Self-Tuning Networks (STNs) [38] incurs little computation and memory overhead, as detailed in section 2.2. Moreover, it is straightforward to implement in deep learning frameworks and is able tune discrete (e.g. number of Cutout holes [9]) and non-differentiable (e.g. dropout rate) hyperparameters. However, the range of applicability to general bilevel problems is slightly more restricted, as hypernetwork approach requires a single inner objective and requires that the outer variables parameterize the training objective (like implicit differentiation but unlike unrolling).

## 5   Experiments

In this section, a series of experiments was conducted to investigate the following questions: (1) How does our method perform in comparison to the STN in terms of convergence, accuracy, and stability? (2) Does our method scale well to modern-size convolutional neural network? (3) Can our method be extended to other architectures such as recurrent neural networks?

We denote our method with the centered parameterization and the modified training objective as "centered STN" (sections 3.1, 3.2), and centered STN with linearization as "$\Delta$-STN" (section 3.3).

### 5.1   Toy Problems

We first validated the $\Delta$-STN on linear regression and linear networks, so that the optimal weights and hyperparameters could be determined exactly. We used regression datasets from the UCI collection [11]. For all experiments, we fixed the perturbation scale to 1, and set $T_{\text{train}} = 10$ and $T_{\text{valid}} = 1$. We compared our method with STNs and the optimal solution to the bilevel problem. We present additional results and a more detailed experimental set-up at appendix H.1.

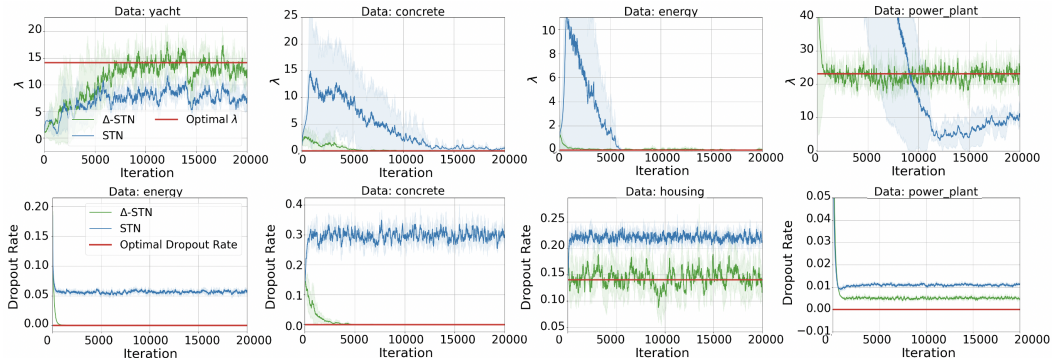

**Figure 3:** A comparison of STNs and Δ-STNs on linear regression tasks (the closer to the optimal, the better). We separately optimize **(top)** weight decay and **(bottom)** input dropout rate. For both tasks, Δ-STNs achieve faster convergence, higher accuracy, and improved stability compared to STNs.

**Linear Regression.** We separately trained linear regression models with $L_2$ regularization and with input dropout. The trajectories for each hyperparameter are shown in figure 3. By reparameterizing the hypernetwork and modifying the training objective, the Δ-STN consistently achieved faster convergence, higher accuracy, and improved stability compared to the STN.

**Linear Networks.** Next, we trained a 5 hidden layer linear network with Jacobian norm regularization. To show the effectiveness of linearization, we present results with and without linearizing the hypernetwork. In figure 4, the centered STN converges to the optimal $\lambda$ more accurately and efficiently than STNs. The linearization further helped improving the accuracy and stability of the approximation.

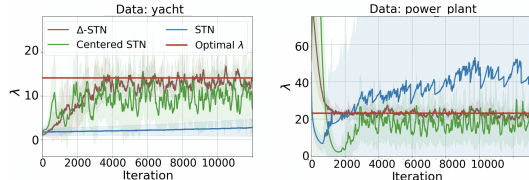

**Figure 4:** A comparison of hyperparameter updates found by STNs, centered STNs, and Δ-STNs on linear network with Jacobian norm regularization.

## 5.2 Image Classification

To evaluate the scalability of our proposed architecture to deep neural networks, we applied Δ-STNs to image classification tasks. We set $T_{\text{train}} = 5$ and $T_{\text{valid}} = 1$ for all experiments and compared Δ-STNs to random search (RS) [2], Bayesian optimization (BO) [53, 54], and Self-Tuning Networks (STNs) [38]. The final performances on validation and test losses are summarized in table 1. Our Δ-STN achieved the best generalization performance for all experiments,

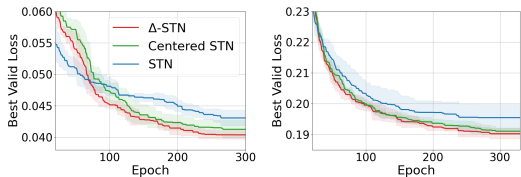

**Figure 5:** A comparison of best validation loss obtained by STNs, centered STNs, and Δ-STNs for **(left)** MNIST and **(right)** FashionMNIST datasets.

demonstrating the effectiveness of our approach. The details of the experiment settings and additional results are provided in appendix H.2. Moreover, we show that Δ-STNs are more robust to hyperparameter initialization and perturbation scale in appendix I.

**MNIST & FashionMNIST.** We applied Δ-STNs on MNIST [29] and FashionMNIST [61] datasets. For MNIST, we trained a multilayer perceptron with 3 hidden layers of rectified units [45, 17]. We tuned 3 dropout rates that control the input and per-layer activations. For FashionMNIST, we trained a convolutional neural network composed of two convolution layers with 32 and 64 filters, followed by 2 fully-connected layers. In total 6 hyperparameters were optimized: input dropout, per-layer dropouts, and Cutout holes and length [9]. Both networks were trained for 300 epochs and a comparison was made between STNs and Δ-STNs in terms of the best validation loss achieved by a given epoch as shown in figure 5. The Δ-STNs was able to achieve a better generalization with faster convergence.

**CIFAR-10.** Finally, we evaluated Δ-STNs on the CIFAR-10 dataset [27]. We used the AlexNet [26], VGG16 [51], and ResNet18 [20] architectures. For all architectures, we tuned (1) input dropout, (2) per-layer dropouts, (3) Cutout holes and length, and (4) amount of noise applied to hue, saturation,

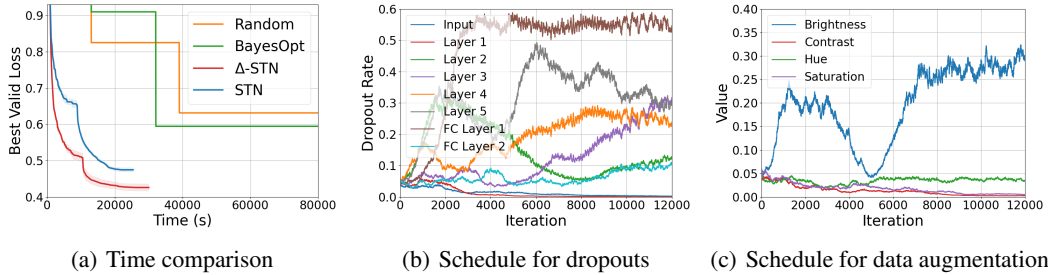

(a) Time comparison     (b) Schedule for dropouts     (c) Schedule for data augmentation

**Figure 6:** **(a)** A comparison of the best validation loss achieved by random search, Bayesian optimization, STNs, and $\Delta$-STNs over time for AlexNet. $\Delta$-STNs achieved the lowest validation loss more efficiently than other methods. **(b)**, **(c)**: Hyperparameter schedules found by $\Delta$-STNs for dropout rates and data augmentation parameters.

**Table 1:** Final validation (test) losses / perplexities of each method on the image classification tasks and the PTB word-level language modeling task.

| Dataset | Network | RS | BO | STN | Centered STN | $\Delta$-STN |
|---------|---------|-----|-----|-----|--------------|--------------|
| MNIST | MLP | 0.043 (0.042) | 0.042 (0.043) | 0.043 (0.041) | 0.041 (0.039) | **0.040** (**0.038**) |
| FMNIST | SimpleCNN | 0.206 (0.214) | 0.217 (0.215) | 0.196 (0.218) | 0.191 (0.212) | **0.189** (**0.209**) |
| CIFAR10 | AlexNet | 0.631 (0.671) | 0.594 (0.598) | 0.474 (0.488) | 0.431 (0.450) | **0.425** (**0.446**) |
|  | VGG16 | 0.566 (0.595) | 0.421 (0.446) | 0.330 (0.354) | 0.286 (0.321) | **0.272** (**0.296**) |
|  | ResNet18 | 0.264 (0.298) | 0.230 (0.267) | 0.266 (0.312) | 0.222 (0.258) | **0.204** (**0.238**) |
| PTB | LSTM | 84.81 (81.46) | 72.13 (69.29) | 70.67 (67.78) | 69.40 (66.67) | **68.63** (**66.26**) |

brightness and contrast to the image, (5) random scale, translation, shear, rotation applied to the image, resulting in total of 18, 26, and 19 hyperparameters. Figure 6 shows the best validation loss achieved by each method over time and the hyperparameter schedules prescribed by $\Delta$-STNs for AlexNet. $\Delta$-STNs achieved the best generalization performance compared to other methods, in less time.

### 5.3 Language Modelling

To demonstrate that $\Delta$-STNs can be extended to different settings, we applied our method to an LSTM network on the Penn Tree Bank (PTB) dataset [40], which has been popularly used for RNN regularization benchmarks because of its small size [16, 43, 60, 38]. We trained a model consisted of 2 LSTM layers and tuned 7 hyperparameters: (1) variational dropout on the input, hidden state, and output, (2) embedding dropout, (3) Drop-Connect [58], and (4) activation regularization coefficients. A more detailed explanation on role of each hyperparameter and the experiment set-up can be found in appendix H.3. The final validation and test perplexities achieved by each method are shown in Table 1. The $\Delta$-STN outperforms other baselines, achieving the lowest validation and test perplexities.

## 6 Conclusion

We introduced $\Delta$-Self-Tuning Networks ($\Delta$-STNs), an improved hypernetwork architecture that efficiently optimizes hyperparameters online. We showed that $\Delta$-STNs fix subtle pathologies in training STNs by (1) reparameterizing the hypernetwork, (2) modifying the training objectives, and (3) linearzing the best-response hypernetwork. The key insight was to accurately approximate the best-response Jacobian instead of the full best-response function. Empirically, we demonstrated that $\Delta$-STNs achieve better generalization in less time, compared to existing approaches to bilevel optimization. We believe that $\Delta$-STNs offer a more reliable, robust, and accurate deployment of hyperparameter optimization based on hypernetwork approach, and offer an alternative method to efficiently solve bilevel optimization problems.

## Acknowledgements and Funding Disclosure

This work was supported by a grant from LG Electronics and a Canada Research Chair. RG acknowledges support from the CIFAR Canadian AI Chairs program. Part of this work was carried out when RG was a visitor at the Institute for Advanced Study in Princeton, NJ. We thank Xuchan Bao, David Duvenaud, Jonathan Lorraine, Matthew MacKay, and Paul Vicol for helpful discussions.

## Broader Impact

Most application of deep learning involves regularization hyperparameters, and hyperparameter tuning is one stage of a much longer pipeline. Hence, any discussion of societal impacts would necessarily be speculative. One predictable impact of this work is to lessen the need for massive computing resources to tune hyperparameters.

## Footnotes

[1]The uniqueness of $\arg \min$ is assumed throughout this paper.

[2]We show the structured hypernetwork representation for convolutional layers in appendix E.

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
