[Supplementary Material]

# A  Table of Notation

| Notation | Description |
|---|---|
| $\boldsymbol{\lambda}, \mathbf{w}$ | Hyperparameters and parameters |
| $\mathcal{L}_T, \mathcal{L}_V$ | Training and validation objectives |
| $h$ | Number of hyperparameters |
| $m$ | Number of parameters |
| $p$ | Number of hypernetwork parameters |
| $n$ | Number of training data |
| $v$ | Number of validation data |
| $\mathbf{r}(\boldsymbol{\lambda})$ | Best-response function |
| $\partial\mathbf{r}/\partial\boldsymbol{\lambda}$ | Best-response Jacobian |
| $\boldsymbol{\epsilon}$ | Perturbation vector |
| $\boldsymbol{\sigma}$ | Perturbation scale |
| $p(\boldsymbol{\epsilon}\|\boldsymbol{\sigma})$ | Gaussian distribution with mean $\mathbf{0}$ and covariance $\mathrm{diag}(\boldsymbol{\sigma})$ |
| $\boldsymbol{\phi}$ | STN's parameters |
| $\mathbf{r}_{\boldsymbol{\phi}}$ | Parametric best-response function with STN's parameterization |
| $\mathbb{H}$ | Entropy term |
| $\tau$ | Entropy weight |
| $\boldsymbol{\theta}$ | $\Delta$-STN's parameters |
| $\mathbf{r}_{\boldsymbol{\theta}}$ | Parametric best-response Jacobian with $\Delta$-STN's parameterization |
| $\mathbf{J_{yw}}$ | Weight-output Jacobian |
| $\mathbf{H}$ | Hessian |
| $\mathbf{G}$ | Gauss-Newton Hessian |
| $\kappa$ | Condition number |
| $\otimes$ | Kronecker product |
| $\alpha$ | Learning rate |
| $\mathbf{y}$ | Network's prediction |
| $f$ | Network function |
| $\odot, \odot_{\mathrm{row}}$ | Element-wise multiplication and row-wise multiplication |
| $T_{\mathrm{train}}, T_{\mathrm{valid}}$ | Training and validation update intervals |
| $\mathbf{X}, \mathbf{t}$ | Training data and target vector |
| $\mathbf{w}^*$ | Optimal parameters |
| $s$ | Fixed non-linear transformation on hyperparameters |

**Table 2:** A summary of notations used in this paper.

# B  Training Algorithm for Self-Tuning Networks (STNs)

In this section, we present the training algorithm for Self-Tuning Networks. The full algorithm is shown in Alg. 2.

---

**Algorithm 2:** Training Algorithm for Self-Tuning Networks (STNs) [38]

---

**Initialize:** Best-response parameters $\phi = \{\phi_0, \mathbf{\Phi}\}$; hyperparameters $\boldsymbol{\lambda}$; learning rates $\{\alpha_i\}_{i=1}^3$;
  training and validation update intervals $T_{\text{train}}$ and $T_{\text{valid}}$; entropy weight $\tau$.
**while** *not converged* **do**
    **for** $t = 1, ..., T_{train}$ **do**
        $\boldsymbol{\epsilon} \sim p(\boldsymbol{\epsilon}|\boldsymbol{\sigma})$
        $\phi \leftarrow \phi - \alpha_1 \nabla_\phi (\mathcal{L}_T(\boldsymbol{\lambda} + \boldsymbol{\epsilon}, r_\phi(\boldsymbol{\lambda} + \boldsymbol{\epsilon})))$
    **for** $t = 1, ..., T_{valid}$ **do**
        $\boldsymbol{\epsilon} \sim p(\boldsymbol{\epsilon}|\boldsymbol{\sigma})$
        $\boldsymbol{\lambda} \leftarrow \boldsymbol{\lambda} - \alpha_2 \nabla_{\boldsymbol{\lambda}}(\mathcal{L}_V(\boldsymbol{\lambda} + \boldsymbol{\epsilon}, r_\phi(\boldsymbol{\lambda} + \boldsymbol{\epsilon})))$
        $\boldsymbol{\sigma} \leftarrow \boldsymbol{\sigma} - \alpha_3 \nabla_{\boldsymbol{\sigma}}(\mathcal{L}_V(\boldsymbol{\lambda} + \boldsymbol{\epsilon}, r_\phi(\boldsymbol{\lambda} + \boldsymbol{\epsilon})) - \tau \mathbb{H}[p(\boldsymbol{\epsilon}|\boldsymbol{\sigma})])$

---

# C  Proofs

## C.1  Observation 1

**Observation 1.** *The Gauss-Newton Hessian with respect to the hypernetwork is given by:*

$$\mathbf{G}_\phi = \mathbb{E}\left[\hat{\boldsymbol{\lambda}}\hat{\boldsymbol{\lambda}}^\top \otimes \mathbf{J}_{\mathbf{yw}}^\top \mathbf{H}_{\mathbf{y}}\mathbf{J}_{\mathbf{yw}}\right], \tag{C.1}$$

*where $\hat{\boldsymbol{\lambda}} = \bar{\boldsymbol{\lambda}} + \bar{\boldsymbol{\epsilon}}$ are the sampled hyperparameters and $\bar{\boldsymbol{\epsilon}}$ is the perturbation vector appended with additional homogeneous coordinate with value 0.*

**Proof.**  The hypernetwork-output Jacobian is as follows:

$$\mathbf{J}_{\mathbf{y}\phi} = \hat{\boldsymbol{\lambda}}^\top \otimes \mathbf{J}_{\mathbf{yw}} \tag{C.2}$$

Then, the Gauss-Newton Hessian with respect to the hypernetwork is:

$$\mathbf{G}_\phi = \mathbb{E}\left[\mathbf{J}_{\mathbf{y}\phi}^\top \mathbf{H}_{\mathbf{y}}\mathbf{J}_{\mathbf{y}\phi}\right] \tag{C.3}$$

$$= \mathbb{E}[(\hat{\boldsymbol{\lambda}} \otimes \mathbf{J}_{\mathbf{yw}}^\top)(\mathbf{1} \otimes \mathbf{H}_{\mathbf{y}})(\hat{\boldsymbol{\lambda}}^\top \otimes \mathbf{J}_{\mathbf{yw}})] \tag{C.4}$$

$$= \mathbb{E}[\hat{\boldsymbol{\lambda}}\hat{\boldsymbol{\lambda}}^\top \otimes \mathbf{J}_{\mathbf{yw}}^\top \mathbf{H}_{\mathbf{y}}\mathbf{J}_{\mathbf{yw}}], \tag{C.5}$$

where $\mathbf{1}$ is a $1 \times 1$ matrix with entry 1.  $\square$

## C.2  Observation 2

**Observation 2.** *Let $\kappa(\mathbf{A})$ denote the condition number of a square positive definite matrix $\mathbf{A}$. Given square positive definite matrices $\mathbf{A}$ and $\mathbf{B}$, the condition number of $\mathbf{A} \otimes \mathbf{B}$ is given by $\kappa(\mathbf{A} \otimes \mathbf{B}) = \kappa(\mathbf{A})\kappa(\mathbf{B})$.*

**Proof.**  Let $\mathbf{A}$ and $\mathbf{B}$ be square positive definite matrices. Then, we get:

$$\kappa(\mathbf{A} \otimes \mathbf{B}) = \|\mathbf{A} \otimes \mathbf{B}\| \, \|\mathbf{A}^{-1} \otimes \mathbf{B}^{-1}\| \tag{C.6}$$

$$= \|\mathbf{A}\| \, \|\mathbf{B}\| \, \|\mathbf{A}^{-1}\| \, \|\mathbf{B}^{-1}\| \tag{C.7}$$

$$= \kappa(\mathbf{A})\kappa(\mathbf{B}) \tag{C.8}$$

$\square$

## C.3 Theorem 3

**Theorem 3.** *Suppose $\mathcal{L}_T$ is quadratic with $\partial^2\mathcal{L}_T/\partial\mathbf{w}^2 \succ 0$ and let $p(\boldsymbol{\epsilon}|\boldsymbol{\sigma})$ be a diagonal Gaussian distribution with mean $\mathbf{0}$ and variance $\sigma^2\mathbf{I}$. Fixing $\boldsymbol{\lambda}_0 \in \mathbb{R}^h$ and $\mathbf{w}_0 \in \mathbb{R}^m$, the solution to the objective in Eqn. 3.10 is the best-response Jacobian.*

**Proof.**

Consider a quadratic function $\mathcal{L}_T \colon \mathbb{R}^h \times \mathbb{R}^m \to \mathbb{R}$ defined as:

$$\mathcal{L}_T(\boldsymbol{\lambda}, \mathbf{w}) = \frac{1}{2}\begin{pmatrix}\mathbf{w}^\top & \boldsymbol{\lambda}^\top\end{pmatrix}\begin{pmatrix}\mathbf{A} & \mathbf{B} \\ \mathbf{B}^\top & \mathbf{C}\end{pmatrix}\begin{pmatrix}\mathbf{w} \\ \boldsymbol{\lambda}\end{pmatrix} + \mathbf{d}^\top\mathbf{w} + \mathbf{e}^\top\boldsymbol{\lambda} + c, \tag{C.9}$$

where $\mathbf{A} \in \mathbb{R}^{m\times m}, \mathbf{B} \in \mathbb{R}^{m\times h}, \mathbf{C} \in \mathbb{R}^{h\times h}, \mathbf{d} \in \mathbb{R}^m, \mathbf{e} \in \mathbb{R}^h$, and $c \in \mathbb{R}$. By assumption, $\mathbf{A}$ is a positive definite matrix. Fixing $\boldsymbol{\lambda}_0 \in \mathbb{R}^h$, the optimal weight is given by:

$$\mathbf{w}^* = -\mathbf{A}^{-1}(\mathbf{B}\boldsymbol{\lambda}_0 + \mathbf{d}) \tag{C.10}$$

and the best-response function is:

$$\mathbf{r}(\boldsymbol{\lambda}) = -\mathbf{A}^{-1}(\mathbf{B}\boldsymbol{\lambda} + \mathbf{d}) \tag{C.11}$$

Consequently, the best-response Jacobian is as follows:

$$\frac{\partial\mathbf{r}}{\partial\boldsymbol{\lambda}}(\boldsymbol{\lambda}_0) = -\mathbf{A}^{-1}\mathbf{B} \in \mathbb{R}^{m\times h} \tag{C.12}$$

Given $\mathbf{w}_0 \in \mathbb{R}^m$, formulating the objective as in Eqn 3.10, we have:

$$\mathbb{E}_{\boldsymbol{\epsilon}\sim p(\boldsymbol{\epsilon}|\boldsymbol{\sigma})}\left[\mathcal{L}_T(\boldsymbol{\lambda}_0 + \boldsymbol{\epsilon}, \mathbf{w}_0 + \boldsymbol{\Theta}\boldsymbol{\epsilon})\right] \tag{C.13}$$

$$= \mathbb{E}_{\boldsymbol{\epsilon}\sim p(\boldsymbol{\epsilon}|\boldsymbol{\sigma})}\left[\frac{1}{2}(\mathbf{w}_0 + \boldsymbol{\Theta}\boldsymbol{\epsilon})^\top\mathbf{A}(\mathbf{w}_0 + \boldsymbol{\Theta}\boldsymbol{\epsilon}) + (\mathbf{w}_0 + \boldsymbol{\Theta}\boldsymbol{\epsilon})^\top\mathbf{B}(\boldsymbol{\lambda}_0 + \boldsymbol{\epsilon})\right. \tag{C.14}$$

$$\left. + \frac{1}{2}(\boldsymbol{\lambda}_0 + \boldsymbol{\epsilon})^\top\mathbf{C}(\boldsymbol{\lambda}_0 + \boldsymbol{\epsilon}) + \mathbf{d}^\top(\mathbf{w}_0 + \boldsymbol{\Theta}\boldsymbol{\epsilon}) + \mathbf{e}^\top(\boldsymbol{\lambda}_0 + \boldsymbol{\epsilon}) + c\right] \tag{C.15}$$

$$= \mathbb{E}_{\boldsymbol{\epsilon}\sim p(\boldsymbol{\epsilon}|\boldsymbol{\sigma})}\left[\textcircled{1} + \textcircled{2} + \textcircled{3} + \textcircled{4}\right], \tag{C.16}$$

where each component is:

$$\textcircled{1} = \frac{1}{2}(\boldsymbol{\epsilon}^\top\boldsymbol{\Theta}^\top\mathbf{A}\boldsymbol{\Theta}\boldsymbol{\epsilon} + 2\boldsymbol{\epsilon}^\top\boldsymbol{\Theta}^\top\mathbf{A}\mathbf{w}_0 + \mathbf{w}_0^\top\mathbf{A}\mathbf{w}_0) \tag{C.17}$$

$$\textcircled{2} = \mathbf{w}_0^\top\mathbf{B}\boldsymbol{\lambda}_0 + \mathbf{w}_0^\top\mathbf{B}\boldsymbol{\epsilon} + \boldsymbol{\epsilon}^\top\boldsymbol{\Theta}^\top\mathbf{B}\boldsymbol{\lambda}_0 + \boldsymbol{\epsilon}^\top\boldsymbol{\Theta}^\top\mathbf{B}\boldsymbol{\epsilon} \tag{C.18}$$

$$\textcircled{3} = \frac{1}{2}(\boldsymbol{\lambda}_0^\top\mathbf{C}\boldsymbol{\lambda}_0 + 2\boldsymbol{\epsilon}^\top\mathbf{C}\boldsymbol{\lambda}_0 + \boldsymbol{\epsilon}^\top\mathbf{C}\boldsymbol{\epsilon}) \tag{C.19}$$

$$\textcircled{4} = \mathbf{d}^\top\mathbf{w}_0 + \mathbf{d}^\top\boldsymbol{\Theta}\boldsymbol{\epsilon} + \mathbf{e}^\top\boldsymbol{\lambda}_0 + \mathbf{e}^\top\boldsymbol{\epsilon} + c \tag{C.20}$$

We simplify these expressions by using linearity of expectation and using the fact that $\mathbb{E}[\boldsymbol{\epsilon}^\top\boldsymbol{\epsilon}] = \sigma^2$:

$$\mathbb{E}_{\boldsymbol{\epsilon}\sim p(\boldsymbol{\epsilon}|\boldsymbol{\sigma})}\left[\textcircled{1}\right] = \frac{1}{2}(\mathrm{Tr}[\boldsymbol{\Theta}^\top\mathbf{A}\boldsymbol{\Theta}]\sigma^2 + \mathbf{w}_0^\top\mathbf{A}\mathbf{w}_0) \tag{C.21}$$

$$\mathbb{E}_{\boldsymbol{\epsilon}\sim p(\boldsymbol{\epsilon}|\boldsymbol{\sigma})}\left[\textcircled{2}\right] = \mathbf{w}_0^\top\mathbf{B}\boldsymbol{\lambda}_0 + \mathrm{Tr}[\boldsymbol{\Theta}^\top\mathbf{B}]\sigma^2 \tag{C.22}$$

$$\mathbb{E}_{\boldsymbol{\epsilon}\sim p(\boldsymbol{\epsilon}|\boldsymbol{\sigma})}\left[\textcircled{3}\right] = \frac{1}{2}(\boldsymbol{\lambda}_0^\top\mathbf{C}\boldsymbol{\lambda}_0 + \mathrm{Tr}[\mathbf{C}]\sigma^2) \tag{C.23}$$

$$\mathbb{E}_{\boldsymbol{\epsilon}\sim p(\boldsymbol{\epsilon}|\boldsymbol{\sigma})}\left[\textcircled{4}\right] = \mathbf{d}^\top\mathbf{w}_0 + \mathbf{e}^\top\boldsymbol{\lambda}_0 + c \tag{C.24}$$

Then, the gradient with respect to $\boldsymbol{\Theta}$ is:

$$\frac{\partial}{\partial\boldsymbol{\Theta}}\left(\mathbb{E}_{\boldsymbol{\epsilon}\sim p(\boldsymbol{\epsilon}|\boldsymbol{\sigma})}\left[\mathcal{L}_T(\boldsymbol{\lambda}_0 + \boldsymbol{\epsilon}, \mathbf{w}_0 + \boldsymbol{\Theta}\boldsymbol{\epsilon})\right]\right) = \mathbf{B}\sigma^2 + \mathbf{A}\boldsymbol{\Theta}\sigma^2 \tag{C.25}$$

Setting the above equation equal to $\mathbf{0}$, the optimal solution $\boldsymbol{\Theta}^*$ is the following:

$$\boldsymbol{\Theta}^* = -\mathbf{A}^{-1}\mathbf{B} = \frac{\partial\mathbf{r}}{\partial\boldsymbol{\lambda}}(\boldsymbol{\lambda}_0), \tag{C.26}$$

which matches the best-response Jacobian. $\qquad\square$

## C.4 Lemma 4

**Lemma 4.** *Let $\boldsymbol{\lambda}_0 \in \mathbb{R}^h$ and choose $\mathbf{w}_0 \in \mathbb{R}^m$ to be the solution to Eqn. 3.9. Suppose $\mathcal{L}_T \in \mathcal{C}^2$ in a neighborhood of $(\boldsymbol{\lambda}_0, \mathbf{w}_0)$ and the Hessian $\partial \mathcal{L}_T / \partial \mathbf{w} (\boldsymbol{\lambda}_0, \mathbf{w}_0)$ is positive definite. Then, for some neighborhood $\mathcal{U}$ of $\boldsymbol{\lambda}_0$, there exists a unique continuously differentiable function $\mathbf{r} \colon \mathcal{U} \to \mathbb{R}^m$ such that $\mathbf{r}(\boldsymbol{\lambda}_0) = \mathbf{w}_0$ and $\partial \mathcal{L}_T / \partial \mathbf{w} (\boldsymbol{\lambda}, \mathbf{r}(\boldsymbol{\lambda})) = \mathbf{0}$ for all $\boldsymbol{\lambda} \in \mathcal{U}$. Moreover, the best-response Jacobian on $\mathcal{U}$ is as follows:*

$$\frac{\partial \mathbf{r}}{\partial \boldsymbol{\lambda}}(\boldsymbol{\lambda}) = -\left[ \frac{\partial^2 \mathcal{L}_T}{\partial \mathbf{w}^2}(\boldsymbol{\lambda}, \mathbf{r}(\boldsymbol{\lambda})) \right]^{-1} \left( \frac{\partial^2 \mathcal{L}_T}{\partial \mathbf{w} \partial \boldsymbol{\lambda}}(\boldsymbol{\lambda}, \mathbf{r}(\boldsymbol{\lambda})) \right) \tag{C.27}$$

**Proof.** Let $\boldsymbol{\lambda}_0 \in \mathbb{R}^h$ and $\mathbf{w}_0 \in \mathbb{R}^m$ be the solution to Eqn. 3.9. Suppose $\mathcal{L}_T$ is $\mathcal{C}^2$ in a neighborhood of $(\boldsymbol{\lambda}_0, \mathbf{w}_0)$. By first-order optimality condition, we have:

$$\frac{\partial \mathcal{L}_T}{\partial \mathbf{w}}(\boldsymbol{\lambda}_0, \mathbf{w}_0) = \mathbf{0} \tag{C.28}$$

Since the Hessian is positive definite, it is invertible, and there exists a unique continuously differentiable function $\mathbf{r} \colon \mathcal{U} \to \mathbb{R}^m$ for some neighborhood $\mathcal{U}$ of $\boldsymbol{\lambda}_0$ such that $\mathbf{r}(\boldsymbol{\lambda}_0) = \mathbf{w}_0$ and:

$$\frac{\partial \mathcal{L}_T}{\partial \mathbf{w}}(\boldsymbol{\lambda}, \mathbf{r}(\boldsymbol{\lambda})) = \mathbf{0} \tag{C.29}$$

for all $\boldsymbol{\lambda} \in \mathcal{U}$ by implicit function theorem. Also, we have:

$$\mathbf{0} = \frac{\mathrm{d}}{\mathrm{d}\boldsymbol{\lambda}} \left( \frac{\partial \mathcal{L}_T}{\partial \mathbf{w}}(\boldsymbol{\lambda}, \mathbf{r}(\boldsymbol{\lambda})) \right) \tag{C.30}$$

$$= \left( \frac{\partial^2 \mathcal{L}_T}{\partial \mathbf{w}^2}(\boldsymbol{\lambda}, \mathbf{r}(\boldsymbol{\lambda})) \frac{\partial \mathbf{r}}{\partial \boldsymbol{\lambda}}(\boldsymbol{\lambda}) + \frac{\partial^2 \mathcal{L}_T}{\partial \mathbf{w} \partial \boldsymbol{\lambda}}(\boldsymbol{\lambda}, \mathbf{r}(\boldsymbol{\lambda})) \right)^{\top} \tag{C.31}$$

for all $\boldsymbol{\lambda} \in \mathcal{U}$. Re-arranging Eqn. C.31, we get:

$$\frac{\partial \mathbf{r}}{\partial \boldsymbol{\lambda}}(\boldsymbol{\lambda}) = -\left[ \frac{\partial^2 \mathcal{L}_T}{\partial \mathbf{w}^2}(\boldsymbol{\lambda}, \mathbf{r}(\boldsymbol{\lambda})) \right]^{-1} \left( \frac{\partial^2 \mathcal{L}_T}{\partial \boldsymbol{\lambda} \partial \mathbf{w}}(\boldsymbol{\lambda}, \mathbf{r}(\boldsymbol{\lambda})) \right) \tag{C.32}$$

$\square$

# D Justification for Linearizing the Best-Response Hypernetwork

Consider the inner-level objective:

$$\mathbf{r}(\boldsymbol{\lambda}) = \underset{\mathbf{w} \in \mathbb{R}^m}{\arg\min} \, \mathbb{E}_{\boldsymbol{\epsilon} \sim p(\boldsymbol{\epsilon}|\boldsymbol{\sigma})} \left[ \mathcal{L}_T(\boldsymbol{\lambda} + \boldsymbol{\epsilon}, \mathbf{w}) \right] \tag{D.1}$$

Let $\mathbf{w}_0 = \mathbf{r}(\boldsymbol{\lambda}_0)$ be the current weights and assume it is the optimal solution. Further assuming we can exchange the integral and the gradient operator, by first-order optimality condition, we get:

$$\frac{\partial}{\partial \mathbf{w}} \mathbb{E}_{\boldsymbol{\epsilon} \sim p(\boldsymbol{\epsilon}|\boldsymbol{\sigma})} \left[ \mathcal{L}_T(\boldsymbol{\lambda}_0 + \boldsymbol{\epsilon}, \mathbf{w}_0) \right] = \mathbb{E}_{\boldsymbol{\epsilon} \sim p(\boldsymbol{\epsilon}|\boldsymbol{\sigma})} \left[ \frac{\partial \mathcal{L}_T}{\partial \mathbf{w}}(\boldsymbol{\lambda}_0 + \boldsymbol{\epsilon}, \mathbf{w}_0) \right] = \mathbf{0} \tag{D.2}$$

Differentiating with respect to $\boldsymbol{\lambda}$, we have:

$$\frac{\mathrm{d}}{\mathrm{d}\boldsymbol{\lambda}} \mathbb{E}_{\boldsymbol{\epsilon} \sim p(\boldsymbol{\epsilon}|\boldsymbol{\sigma})} \left[ \frac{\partial \mathcal{L}_T}{\partial \mathbf{w}}(\boldsymbol{\lambda} + \boldsymbol{\epsilon}, \mathbf{r}(\boldsymbol{\lambda})) \right] = \mathbb{E}_{\boldsymbol{\epsilon} \sim p(\boldsymbol{\epsilon}|\boldsymbol{\sigma})} \left[ \frac{\mathrm{d}}{\mathrm{d}\boldsymbol{\lambda}} \left( \frac{\partial \mathcal{L}_T}{\partial \mathbf{w}}(\boldsymbol{\lambda} + \boldsymbol{\epsilon}, \mathbf{r}(\boldsymbol{\lambda})) \right) \right] = \mathbf{0} \tag{D.3}$$

Then:

$$\mathbb{E}_{\boldsymbol{\epsilon} \sim p(\boldsymbol{\epsilon}|\boldsymbol{\sigma})} \left[ \frac{\partial^2 \mathcal{L}_T}{\partial \mathbf{w} \partial \boldsymbol{\lambda}}(\boldsymbol{\lambda}_0 + \boldsymbol{\epsilon}, \mathbf{w}_0) + \frac{\partial^2 \mathcal{L}_T}{\partial \mathbf{w}^2}(\boldsymbol{\lambda}_0 + \boldsymbol{\epsilon}, \mathbf{w}_0) \frac{\partial \mathbf{r}}{\partial \boldsymbol{\lambda}}(\boldsymbol{\lambda}_0) \right] = \mathbf{0} \tag{D.4}$$

For simplicity, we denote:

$$\mathbf{B}(\boldsymbol{\lambda}_0, \mathbf{w}_0, \boldsymbol{\epsilon}) = \frac{\partial^2 \mathcal{L}_T}{\partial \mathbf{w} \partial \boldsymbol{\lambda}}(\boldsymbol{\lambda}_0 + \boldsymbol{\epsilon}, \mathbf{w}_0) \in \mathbb{R}^{m \times h} \tag{D.5}$$

$$\mathbf{A}(\boldsymbol{\lambda}_0, \mathbf{w}_0, \boldsymbol{\epsilon}) = \frac{\partial^2 \mathcal{L}_T}{\partial \mathbf{w}^2}(\boldsymbol{\lambda}_0 + \boldsymbol{\epsilon}, \mathbf{w}_0) \in \mathbb{R}^{m \times m} \tag{D.6}$$

$$\boldsymbol{\Theta} = \frac{\partial \mathbf{r}}{\partial \boldsymbol{\lambda}}(\boldsymbol{\lambda}_0) \in \mathbb{R}^{m \times h} \tag{D.7}$$

Thus, $\boldsymbol{\Theta}$ is the best-response Jacobian, and it is given by:

$$\boldsymbol{\Theta} = -\mathbb{E}[\mathbf{A}]^{-1}\mathbb{E}[\mathbf{B}] \tag{D.8}$$

We can represent the solution as a minimization problem:

$$\boldsymbol{\Theta}^* = \underset{\boldsymbol{\Theta} \in \mathbb{R}^{m \times h}}{\arg\min} \, \mathbb{E}_{\boldsymbol{\epsilon} \sim p(\boldsymbol{\epsilon}|\boldsymbol{\sigma})} \left[ \frac{1}{2}\mathrm{tr}[\mathbf{A}\boldsymbol{\Theta}\boldsymbol{\Theta}^\top] + \mathrm{tr}[\mathbf{B}^\top\boldsymbol{\Theta}] \right] \tag{D.9}$$

The first term in Eqn. D.4 can be represented as:

$$\mathbb{E}_{\boldsymbol{\epsilon} \sim p(\boldsymbol{\epsilon}|\boldsymbol{\sigma})}\left[ \mathbf{B}(\boldsymbol{\lambda}_0, \mathbf{w}_0, \boldsymbol{\epsilon}) \right] = \mathbb{E}_{\boldsymbol{\epsilon} \sim p(\boldsymbol{\epsilon}|\boldsymbol{\sigma})}\left[ \frac{\partial^2 \mathcal{L}_T}{\partial \mathbf{w} \partial \boldsymbol{\lambda}}(\boldsymbol{\lambda}_0 + \boldsymbol{\epsilon}, \mathbf{w}_0) \right] \tag{D.10}$$

$$= \mathbb{E}_{\tilde{\boldsymbol{\epsilon}} \sim p(\tilde{\boldsymbol{\epsilon}}|\mathbf{I})}\left[ \frac{\partial^2 \tilde{\mathcal{L}}_T}{\partial \mathbf{w} \partial \tilde{\boldsymbol{\epsilon}}}(\tilde{\boldsymbol{\epsilon}}, \mathbf{w}_0)\boldsymbol{\Sigma}^{-1/2} \right] \tag{D.11}$$

$$= \mathbb{E}_{\tilde{\boldsymbol{\epsilon}} \sim p(\tilde{\boldsymbol{\epsilon}}|\mathbf{I})}\left[ \left( \frac{\partial \tilde{\mathcal{L}}_T}{\partial \mathbf{w}}(\tilde{\boldsymbol{\epsilon}}, \mathbf{w}_0) \right)^\top \tilde{\boldsymbol{\epsilon}}^\top \boldsymbol{\Sigma}^{-1/2} \right] \tag{D.12}$$

$$= \mathbb{E}_{\boldsymbol{\epsilon} \sim p(\boldsymbol{\epsilon}|\boldsymbol{\sigma})}\left[ \left( \frac{\partial \mathcal{L}_T}{\partial \mathbf{w}}(\boldsymbol{\lambda}_0 + \boldsymbol{\epsilon}, \mathbf{w}_0) \right)^\top \boldsymbol{\epsilon}^\top \boldsymbol{\Sigma}^{-1} \right], \tag{D.13}$$

where $\boldsymbol{\epsilon} = \boldsymbol{\Sigma}^{1/2}\tilde{\boldsymbol{\epsilon}}$ and $\mathcal{L}_T(\boldsymbol{\lambda}_0 + \boldsymbol{\Sigma}^{1/2}\tilde{\boldsymbol{\epsilon}}, \mathbf{w}) = \tilde{\mathcal{L}}_T(\tilde{\boldsymbol{\epsilon}}, \mathbf{w})$. The third line (Eqn. D.12) uses Stein's identity. Multipying $\boldsymbol{\Sigma}$ in Eqn. D.4, we have:

$$\mathbb{E}_{\boldsymbol{\epsilon} \sim p(\boldsymbol{\epsilon}|\boldsymbol{\sigma})}\left[ \mathbf{B}(\boldsymbol{\lambda}_0, \mathbf{w}_0, \boldsymbol{\epsilon})\boldsymbol{\Sigma} + \mathbf{A}(\boldsymbol{\lambda}_0, \mathbf{w}_0, \boldsymbol{\epsilon})\boldsymbol{\Theta}\boldsymbol{\Sigma} \right] = \mathbf{0} \tag{D.14}$$

with the optimization problem:

$$\boldsymbol{\Theta}^* = \underset{\boldsymbol{\Theta} \in \mathbb{R}^{m \times h}}{\arg\min} \, \mathbb{E}_{\boldsymbol{\epsilon} \sim p(\boldsymbol{\epsilon}|\boldsymbol{\sigma})} \left[ \frac{1}{2}\mathrm{tr}[\mathbf{A}\boldsymbol{\Theta}\boldsymbol{\Sigma}\boldsymbol{\Theta}^\top] + \mathrm{tr}[\mathbf{B}^\top\boldsymbol{\Theta}\boldsymbol{\Sigma}] \right] \tag{D.15}$$

The second term in Eqn D.15 is:

$$\mathbb{E}_{\boldsymbol{\epsilon} \sim p(\boldsymbol{\epsilon}|\boldsymbol{\sigma})}\left[ \mathrm{tr}[\mathbf{B}^\top\boldsymbol{\Theta}\boldsymbol{\Sigma}] \right] = \mathbb{E}_{\boldsymbol{\epsilon} \sim p(\boldsymbol{\epsilon}|\boldsymbol{\sigma})}\left[ \mathrm{tr}[\boldsymbol{\Sigma}\mathbf{B}^\top\boldsymbol{\Theta}] \right] \tag{D.16}$$

$$= \mathbb{E}_{\boldsymbol{\epsilon} \sim p(\boldsymbol{\epsilon}|\boldsymbol{\sigma})}\left[ \mathrm{tr}\left[ \boldsymbol{\Sigma}\boldsymbol{\Sigma}^{-1}\boldsymbol{\epsilon}\frac{\partial \mathcal{L}_T}{\partial \mathbf{w}}(\boldsymbol{\lambda}_0 + \boldsymbol{\epsilon}, \mathbf{w}_0)\boldsymbol{\Theta} \right] \right] \tag{D.17}$$

$$= \mathbb{E}_{\boldsymbol{\epsilon} \sim p(\boldsymbol{\epsilon}|\boldsymbol{\sigma})}\left[ \mathrm{tr}\left[ \frac{\partial \mathcal{L}_T}{\partial \mathbf{w}}(\boldsymbol{\lambda}_0 + \boldsymbol{\epsilon}, \mathbf{w}_0)\boldsymbol{\Theta}\boldsymbol{\epsilon} \right] \right] \tag{D.18}$$

$$= \mathbb{E}_{\boldsymbol{\epsilon} \sim p(\boldsymbol{\epsilon}|\boldsymbol{\sigma})}\left[ \mathrm{tr}\left[ \frac{\partial \mathcal{L}_T}{\partial \mathbf{w}}(\boldsymbol{\lambda}_0 + \boldsymbol{\epsilon}, \mathbf{w}_0)\Delta\mathbf{w} \right] \right] \tag{D.19}$$

$$= \mathbb{E}_{\boldsymbol{\epsilon} \sim p(\boldsymbol{\epsilon}|\boldsymbol{\sigma})}\left[ \mathrm{tr}\left[ \frac{\partial \mathcal{L}_T}{\partial \mathbf{y}}(\boldsymbol{\lambda}_0 + \boldsymbol{\epsilon}, \mathbf{w}_0)\Delta\mathbf{y} \right] \right], \tag{D.20}$$

where $\Delta\mathbf{w} = \boldsymbol{\Theta}\boldsymbol{\epsilon}$ and $\Delta\mathbf{y} = \mathbf{J_{yw}}\Delta\mathbf{w}$. On the other hand, the first term is:

$$\mathbb{E}_{\boldsymbol{\epsilon}\sim p(\boldsymbol{\epsilon}|\boldsymbol{\sigma})}\left[\mathrm{tr}[\mathbf{A}(\boldsymbol{\lambda}_0, \mathbf{w}_0, \boldsymbol{\epsilon})\boldsymbol{\Theta}\boldsymbol{\Sigma}\boldsymbol{\Theta}^\top]]\right] \tag{D.21}$$

$$= \mathbb{E}_{\boldsymbol{\epsilon}\sim p(\boldsymbol{\epsilon}|\boldsymbol{\sigma})}\left[\mathrm{tr}\left[\frac{\partial^2 \mathcal{L}_T}{\partial\mathbf{w}^2}(\boldsymbol{\lambda}_0 + \boldsymbol{\epsilon}, \mathbf{w}_0)\boldsymbol{\Theta}\boldsymbol{\Sigma}\boldsymbol{\Theta}^\top\right]\right] \tag{D.22}$$

$$\approx \mathbb{E}_{\boldsymbol{\epsilon}\sim p(\boldsymbol{\epsilon}|\boldsymbol{\sigma})}\left[\mathrm{tr}\left[\frac{\partial^2 \mathcal{L}_T}{\partial\mathbf{w}^2}(\boldsymbol{\lambda}_0 + \boldsymbol{\epsilon}, \mathbf{w}_0)\boldsymbol{\Theta}\boldsymbol{\epsilon}\boldsymbol{\epsilon}^\top\boldsymbol{\Theta}^\top\right]\right] \tag{D.23}$$

$$= \mathbb{E}_{\boldsymbol{\epsilon}\sim p(\boldsymbol{\epsilon}|\boldsymbol{\sigma})}\left[\mathrm{tr}\left[\frac{\partial^2 \mathcal{L}_T}{\partial\mathbf{w}^2}(\boldsymbol{\lambda}_0 + \boldsymbol{\epsilon}, \mathbf{w}_0)\Delta\mathbf{w}\Delta\mathbf{w}^\top\right]\right] \tag{D.24}$$

$$= \mathbb{E}_{\boldsymbol{\epsilon}\sim p(\boldsymbol{\epsilon}|\boldsymbol{\sigma})}\left[\mathrm{tr}\left[\Delta\mathbf{w}^\top\frac{\partial^2 \mathcal{L}_T}{\partial\mathbf{w}^2}(\boldsymbol{\lambda}_0 + \boldsymbol{\epsilon}, \mathbf{w}_0)\Delta\mathbf{w}\right]\right] \tag{D.25}$$

$$\approx \mathbb{E}_{\boldsymbol{\epsilon}\sim p(\boldsymbol{\epsilon}|\boldsymbol{\sigma})}\left[\mathrm{tr}\left[\Delta\mathbf{w}^\top\mathbf{J_{yw}}^\top\frac{\partial^2 \mathcal{L}_T}{\partial\mathbf{y}^2}(\boldsymbol{\lambda}_0 + \boldsymbol{\epsilon}, \mathbf{w}_0)\mathbf{J_{yw}}\Delta\mathbf{w}\right]\right] \tag{D.26}$$

$$= \mathbb{E}_{\boldsymbol{\epsilon}\sim p(\boldsymbol{\epsilon}|\boldsymbol{\sigma})}\left[\mathrm{tr}\left[\Delta\mathbf{y}^\top\frac{\partial^2 \mathcal{L}_T}{\partial\mathbf{y}^2}(\boldsymbol{\lambda}_0 + \boldsymbol{\epsilon}, \mathbf{w}_0)\Delta\mathbf{y}\right]\right] \tag{D.27}$$

The third line (Eqn. D.23) assumes that $\partial^2\mathcal{L}_T/\partial\mathbf{w}^2(\boldsymbol{\lambda}_0 + \boldsymbol{\epsilon}, \mathbf{w}_0)$ and $\boldsymbol{\epsilon}$ are independent, and Eqn. D.26 is a Gauss-Newton approximation. Therefore, first and second terms correspond to the first- and second-order Taylor approximations to the loss. For $\Delta$-STNs, we linearized the predictions with respect to the loss. This can be explained by the fact that the loss functions such as mean squared error and cross entropy are locally quadratic and closely matches the second order approximation.

## E   Structured Hypernetwork Representation for Convolutional Layers

In this section, we describe a structured best-response approximation for convolutional layers. Considering $i$-th layer of a convolutional neural network, let $C_i$ denote number of filters and $K_i$ denote size of the kernel. Let $\mathbf{W}^{(i,c)} \in \mathbb{R}^{C_{i-1}\times K_i\times K_i}$ and $\mathbf{b}^{(i,c)} \in \mathbb{R}$ denote weights and bias at $c$-th convolutional kernel, where $c \in \{1, ..., C_i\}$. We propose to approximate the layer-wise best-response function as follows:

$$\begin{aligned}
\mathbf{W}_{\boldsymbol{\theta}}^{(i,c)}(\boldsymbol{\lambda}, \boldsymbol{\lambda}_0) &= \mathbf{W}_{\text{general}}^{(i,c)} + \left((\boldsymbol{\lambda} - \boldsymbol{\lambda}_0)^\top\mathbf{u}^{(i,c)}\right) \odot \mathbf{W}_{\text{response}}^{(i,c)}\\
\mathbf{b}_{\boldsymbol{\theta}}^{(i,c)}(\boldsymbol{\lambda}, \boldsymbol{\lambda}_0) &= \mathbf{b}_{\text{general}}^{(i,c)} + \left((\boldsymbol{\lambda} - \boldsymbol{\lambda}_0)^\top\mathbf{v}^{(i,c)}\right) \odot \mathbf{b}_{\text{response}}^{(i,c)},
\end{aligned} \tag{E.1}$$

where $\mathbf{u}^{(i,c)}, \mathbf{v}^{(i,c)} \in \mathbb{R}^h$. Observe that these formulas are linear in $\boldsymbol{\lambda}$ similar to the approximation for fully-connected layers and analogous to that of the original STN. This architecture is also memory efficient and tractable to compute, and allows parallelism. The approximation requires $2|\mathbf{W}^{(i,c)}| + h$ and $2|\mathbf{b}^{(i,c)}| + h$ parameters to represent the weight and bias, and two additional element-wise multiplications in the forward pass. Summing over all channels, the total number of parameters is $2p + 2hC_i$ for each layer, where $p$ is the number parameters for the ordinary CNN layer. Thus, the $\Delta$-STN incurs little memory overhead compared to training an ordinary CNN.

## F   Fixed Nonlinear Function on Hyperparameters

In general, the hyperparameters have restricted domains. For example, the weight decay has to be a positive real number and the dropout rate has to be in between 0 and 1. Hence, we apply a fixed non-linear function $s\colon \mathbb{R}^h \to \mathbb{R}^h$ on the hyperparameters to ensure that hyperparameters are in its domain and optimize the hyperparameters on an unrestricted domain. Fixing $\boldsymbol{\lambda}_0 \in \mathbb{R}^h$, the training objective for the hypernetwork with a fixed nonlinear transformation on the hyperparameters is as follows:

$$\min_{\boldsymbol{\Theta}\in\mathbb{R}^{m\times h}} \mathbb{E}_{\boldsymbol{\epsilon}\sim p(\boldsymbol{\epsilon}|\boldsymbol{\sigma})}\left[\mathcal{L}_T(s(\boldsymbol{\lambda}_0 + \boldsymbol{\epsilon}), \mathbf{r}_{\boldsymbol{\theta}}(\boldsymbol{\lambda}_0 + \boldsymbol{\epsilon}, \boldsymbol{\lambda}_0))\right] \tag{F.1}$$

We also use such transformation to restrict the hyperparameters to be in its search space.

# G  Example of STN's Training Objective having Incorrect Fixed Point

Consider a linear regression with $L_2$ regularization where the training objective is defined as:

$$\mathcal{L}_T(\lambda, \mathbf{w}) = \frac{1}{2n} \|\mathbf{X}\mathbf{w} - \mathbf{t}\|^2 + \frac{\lambda}{2n} \|\mathbf{w}\|^2, \tag{G.1}$$

where $\mathbf{X} \in \mathbb{R}^{n \times m}$ and $\mathbf{t} \in \mathbb{R}^n$ are the input matrix and target vector, respectively. Given $\lambda_0 \in \mathbb{R}$, under the STN's training objective (Eqn. 2.5), we aim to minimize:

$$\min_{\mathbf{w}_0 \in \mathbb{R}^m} \mathbb{E}_{\epsilon \sim p(\epsilon|\sigma)} \left[ \frac{1}{2n} \|\mathbf{X}(\mathbf{w}_0 + \boldsymbol{\Theta}\epsilon) - \mathbf{t}\|^2 + \frac{\lambda_0 + \epsilon}{2n} \|\mathbf{w}_0 + \boldsymbol{\Theta}\epsilon\|^2 \right] \tag{G.2}$$

Simplifying the above equation, we get:

$$\mathbb{E}_{\boldsymbol{\epsilon} \sim p(\boldsymbol{\epsilon}|\boldsymbol{\sigma})} \left[ \textcircled{1} + \textcircled{2} \right], \tag{G.3}$$

where each component is:

$$\mathbb{E}_{\epsilon \sim p(\epsilon,\sigma)} \left[ \textcircled{1} \right] = \frac{1}{2n} \left( \mathbf{w}_0^\top \mathbf{X}^\top \mathbf{X} \mathbf{w}_0 + \boldsymbol{\Theta}^\top \mathbf{X}^\top \mathbf{X} \boldsymbol{\Theta} \sigma^2 + \mathbf{t}^\top \mathbf{t} - 2\mathbf{w}_0^\top \mathbf{X}^\top \mathbf{t} \right) \tag{G.4}$$

$$\mathbb{E}_{\epsilon \sim p(\epsilon,\sigma)} \left[ \textcircled{2} \right] = \frac{1}{2n} \left( \mathbf{w}_0^\top \mathbf{w}_0 \lambda_0 + 2\mathbf{w}_0^\top \boldsymbol{\Theta} \sigma^2 + \boldsymbol{\Theta}^\top \boldsymbol{\Theta} \lambda_0 \sigma^2 \right) \tag{G.5}$$

The gradient with respect to $\mathbf{w}_0$ is the following:

$$\frac{\partial}{\partial \mathbf{w}_0} \mathbb{E}_{\epsilon \sim p(\epsilon|\sigma)} \left[ \mathcal{L}_T(\lambda_0 + \epsilon, \mathbf{r}_{\boldsymbol{\theta}}(\lambda_0 + \epsilon, \lambda_0)) \right] = \frac{1}{n} \left( \mathbf{X}^\top \mathbf{X} \mathbf{w}_0 - \mathbf{X}^\top \mathbf{t} + \mathbf{w}_0 \lambda_0 + \boldsymbol{\Theta} \sigma^2 \right) \tag{G.6}$$

Setting the above equation equal to $\mathbf{0}$, the optimal solution $\mathbf{w}_0^*$ under STN's training objective is as follows:

$$\mathbf{w}_0^* = (\mathbf{X}^\top \mathbf{X} + \lambda_0 \mathbf{I})^{-1} (\mathbf{X}^\top \mathbf{t} - \boldsymbol{\Theta} \sigma^2) \tag{G.7}$$

and the optimal $\boldsymbol{\Theta}^*$ is:

$$\boldsymbol{\Theta}^* = -(\mathbf{X}^\top \mathbf{X} + \lambda_0 \mathbf{I})^{-1} \mathbf{w}_0^*, \tag{G.8}$$

Comparing Eqn. G.7 to the optimal weight $\mathbf{w}^*$ of linear regression with $L_2$ regularization,

$$\mathbf{w}^* = (\mathbf{X}^\top \mathbf{X} + \lambda_0 \mathbf{I})^{-1} \mathbf{X}^\top \mathbf{t}, \tag{G.9}$$

the optimal solution for $\mathbf{w}_0^*$ under STN's training objective is incorrect, as $\sigma^2 > 0$. Moreover, the inaccuracy of $\mathbf{w}_0^*$ also affects accuracy of the best-response Jacobian as shown in Eqn. G.8. In contrast, the proposed objective in Eqn. 3.9 recovers the correct solution for both the weight $\mathbf{w}_0^*$ and best-response Jacobian $\boldsymbol{\Theta}^*$.

# H  Experiment Details

In this section, we present additional details for each experiment.

## H.1  Toy Experiments

For toy experiments, the datasets were randomly split into training and validation set with ratio 80% and 20%. We fixed the perturbation scale to 1, and used $T_{\text{train}} = 10$ and $T_{\text{valid}} = 1$ for all experiments. We further used 100 iterations of warm-up that does not update the hyperparameters. Note that the warm-up still perturbs the hyperparameters. The batch size was 10 for datasets with less than 1000 data points, and 100 for others. In training, we normalized the input features and targets to have a zero mean and unit variance. The training objective for the ridge regression was as follows:

$$\mathcal{L}_T(\lambda, \mathbf{w}) = \frac{1}{2n} \|\mathbf{X}\mathbf{w} - \mathbf{t}\|^2 + \frac{\exp(\lambda)}{2n} \|\mathbf{w}\|^2, \tag{H.1}$$

where $n$ is the number of training data, $\mathbf{X} \in \mathbb{R}^{n \times m}$ is the input matrix and $\mathbf{t} \in \mathbb{R}^n$ is the target. We initialized the regularization penalty $\lambda$ to 1 for datasets with less than 1000 data points and 50 for others. The hyperparameter updates for linear regression with $L_2$ regularization on UCI datasets is shown in figure 7.

We note that dividing the regularization penalty by the number of training set as in Eqn. H.1 is non-standard. Because the STN is sensitive to the scale of the hyperparameters, we also experimented with the objective that does not divide the regularization penalty by the number of training data. As shown in figure 8, $\Delta$-STNs still outperform STNs in terms of convergence, accuracy, and stability.

**Figure 7:** A comparison of STNs and $\Delta$-STNs on linear regression with $L_2$ regularization.

**Figure 8:** A comparison of STNs and $\Delta$-STNs on linear regression with $L_2$ regularization without dividing the regularization penalty by the number of training data.

In figure 9, we decoupled the centered parameterization from all other changes to the training objective for datasets that experimented spikes in the hyperparameter updates. The centering ("Reparam STN") described in section 3.1 by itself eliminated the spike in the hyperparameter updates as our analysis predicts. The modification of the training objective ("Reparam STN + New Obj.") described in section 3.2 further helped improving the accuracy of the fixed point as detailed in appendix G.

**Figure 9:** A comparison of hyperparameter updates found by STNs and $\Delta$-STNs, decoupling effects of reparameterization and modified objective.

Similarly, the training objective for dropout experiments was:

$$\mathcal{L}_T(\lambda, \mathbf{w}) = \frac{1}{2n} \left\| (\mathbf{R} \odot \mathbf{X})\mathbf{w} - \mathbf{t} \right\|^2, \tag{H.2}$$

where $\mathbf{R} \in \mathbb{R}^{n \times m}$ is a random matrix with $\mathbf{R}_{ij} \sim \text{Bern}(\lambda)$ and $\odot$ is an element-wise multiplication. Note that linear regression with input dropout is, in expectation, equivalent to ridge regression with a particular form of regularization penalty [55]. We initialized the dropout rate to 0.2 for all datasets. For both experiments, the validation objective was the following:

$$\mathcal{L}_V(\lambda, \mathbf{w}) = \frac{1}{2v} \left\| \mathbf{X}_{\text{valid}}\mathbf{w} - \mathbf{t}_{\text{valid}} \right\|^2, \tag{H.3}$$

where $v$ is the number of validation data, and $\mathbf{X}_{\text{valid}} \in \mathbb{R}^{v \times m}$ and $\mathbf{t}_{\text{valid}} \in \mathbb{R}^{v}$ are the inputs and targets of the validation data.

We adopted the same experimental settings for linear networks with Jacobian norm regularization. The model was consisted of 5 hidden layers and each hidden layer had a square matrix with the same dimension as the input. The training objective was as follows:

$$\mathcal{L}_T(\lambda, \mathbf{w}) = \frac{1}{2n} \|y(\mathbf{X}; \mathbf{w}) - \mathbf{t}\|^2 + \frac{\exp(\lambda)}{2n} \left\| \frac{\partial y}{\partial \mathbf{X}}(\mathbf{X}; \mathbf{w}) \right\|^2, \tag{H.4}$$

where $y$ is the linear network's prediction. The plots presented in figures 3 and 4 show the updates of hyperparameters averaged over 5 runs with different random seeds.

## H.2 Image Classification

For the baselines (grid search, random search, and Bayesian optimization), the search spaces were as follows: dropout rates were in $[0, 0.85]$; the number of Cutout holes was in $[0, 4]$; Cutout length was in $[0, 24]$, noises added to contrast, brightness, and saturation were in $[0, 1]$; noise added to hue was in $[0, 0.5]$; the random scale and translation of an image were in $[0, 0.5]$; the random degree and shear were in $[0, 45]$. All other settings (e.g. learning rate schedule, batch size) were the same as those of $\Delta$-STNs. We used Tune [33] for gird and random searches, and Spearmint [53] for Bayesian optimization.

For all experiments, we set training and validation update intervals to $T_{\text{train}} = 5$ and $T_{\text{valid}} = 1$, and the perturbation scale was initialized to 1. We further performed a grid search over the entropy weight from a set $\{1e^{-2}, 1e^{-3}, 1e^{-4}\}$ on 1 run and repeated the experiments with the selected entropy weight 5 times. We reported the average over 5 runs.

### H.2.1 MNIST

We held out 15% of the training data for validation. We trained a multilayer perceptron using SGD with a fixed learning rate 0.01 and momentum 0.9, and mini-batches of size 128 for 300 epochs. The MLP model consisted of 3 hidden layers with 1200 units and ReLU activations. We used 5 warm-up epochs that did not optimize the hyperparameters. In total of 3 dropout rates that control the inputs and per-layer activations were tuned. The hyperparameters were optimized using RMSProp with learning rate 0.01. We used entropy weights $\tau = 0.001$ for both STNs and $\Delta$-STNs, and initialized all dropout rates to 0.05, where the range for dropout was $[0, 0.95]$. We show the hyperparameter schedules found by $\Delta$-STNs on figure 10.

### H.2.2 FashionMNIST

Similar to the MNIST experiment, we held out 15% of the training data for validation. We trained Convolutional neural network using SGD with a fixed learning rate 0.01 and momentum 0.9, and mini-batches of size 128 for 300 epochs. The SimpleCNN model consisted of 2 convolutional layers with 16 and 32 filters, both with kernel size 5, followed by 2 fully-connected layers with 1568 hidden units and with ReLU activations on all hidden layers. We tuned 6 hyperparameters: (1) input dropout, (2) per-layer activation dropouts, and (3) Cutout holes and length. We set 5 warm-up epochs that did not optimize the hyperparameters and used RMSProp with learning rate 0.01 for optimizing the hyperparameters. The entropy weight was $\tau = 0.001$ for both STNs and $\Delta$-STNs. We initialized the dropout rates to $0.05$, the number of Cutout holes to 1, and Cutout length to 4. Dropout rates had a search space of $[0, 0.95]$, the number of Cutout holes had $[0, 4]$, and Cutout length had $[0, 24]$. The hyperparameter schedules prescribed by $\Delta$-STNs are shown in figure 10.

### H.2.3 CIFAR10

We held out 20% of the training data for validation. AlexNet, VGG16, and ResNet18 were trained with SGD with initial learning rates of 0.03, 0.01, 0.05 and momentum 0.9, using mini-batches of size 128. We decayed the learning rate after 60 epochs by a factor of 0.2 and trained the network for 200 epochs. We tuned 18 and 26 hyperparameters for AlexNet and VGG16: (1) input dropout, (2) per-layer activation dropouts, (3) scaling noise applied to the input, (4) Cutout holes and length, (5) amount of noise applied to hue, saturation, brightness, and contrast to the image, and (6)

(a) MLP dropout     (b) SimpleCNN dropout     (c) SimpleCNN Cutout

**Figure 10:** Hyperparameter schedules prescribed by $\Delta$-STNs for **(a)** MNIST and **(b)**, **(c)** FashionMNIST datasets.

(a) Schedule for dropouts     (b) Schedule for data augmentations     (c) Schedule for Cutout

**Figure 11:** Hyperparameter schedules found by $\Delta$-STNs on VGG16 for **(a)** dropout rates, **(b)** data augmentation parameters, and **(c)** Cutout parameters.

random translation, scale, rotation, and shear applied to input image. Similarly, 19 hyperparameters were optimized for ResNet18, where we applied dropout after each block and applied the same augmentation hyperparameters as AlexNet and VGG16.

**Table 3:** Final validation (test) accuracy of STN and $\Delta$-STN on image classification tasks.

| Network | STN | $\Delta$-STN |
|---|---|---|
| AlexNet | 83.96 (83.38) | **85.63 (85.19)** |
| VGG16 | 89.22 (88.66) | **90.96 (90.26)** |
| ResNet18 | 91.51 (90.16) | **93.46 (92.55)** |

The hyperparameters were optimized using RMSProp with a fixed learning rate 0.01. For all STN-type models, we used 5 epochs of warm-up for the model parameters. We used entropy weights of $\tau = 0.001$ on AlexNet, and $\tau = 0.0001$ and $\tau = 0.001$ on VGG16 for $\Delta$-STNs and STNs, respectively. For ResNet18, we used entropy weight of $\tau = 0.0001$. The number Cutout holes was initialized to 1, the Cutout length was initialized to 4, and all other hyperparameters were initialized to 0.05. The search spaces for amount of noises added to contrast, brightness, saturation were $[0, 1]$ and that for noise added hue was $[0, 0.5]$. The search spaces for random translation and scale were also $[0, 0.5]$ and those for random rotation and shear were $[0, 45]$. The search spaces for all other hyperparameters were the same as those in experiments for FashionMNIST. We show the hyperparameter schedules obtained by $\Delta$-STNs on figure 11 and figure 12 for VGG16 and ResNet18 architectures. We also show the validation (training) accuracy obtained by each architecture in table 3. $\Delta$-STNs showed a consistent improvement in validation accuracy compared to STNs.

### H.3 Language Modeling

We adopted the same experiment set-up to that of MacKay et al. [38]. We trained a 2 layer LSTM with 650 hidden units and 650-dimensional word embedding on sequences of length 70, using mini-batches of size 40. We used SGD with initial learning rate of 30 and decayed by a factor of 4 when the validation loss did not improve for 5 epochs. We further used gradient clipping with parameter 0.25. The

**Figure 13:** Hyperparameter schedules prescribed by $\Delta$-STNs on LSTM experiments.

hyperparameters were optimized using Adam with a fixed learning rate of 0.01. We used 10 warm-up

(a) Schedule for dropouts    (b) Schedule for data augmentations    (c) Schedule for Cutout

**Figure 12:** Hyperparameter schedules found by $\Delta$-STNs on ResNet18 for **(a)** dropout rates, **(b)** data augmentation parameters, and **(c)** Cutout parameters.

(a) Schedule found by $\Delta$-STNs    (b) Schedule found by STNs

**Figure 14:** A comparison of input dropout schedules found by **(a)** $\Delta$-STNs and **(b)** STNs on MLP with different initialization. $\Delta$-STNs found the hyperparameter schedule more robustly and accurately compared to STNs.

epochs for both STNs and $\Delta$-STNs with a fixed perturbation scale of 1 and terminated the training when the learning rate for hypernetwork parameters decreased below 0.0003.

In total of 7 hyperparameters were optimized. We tuned variational dropout applied to the inputs, hidden states between layers, and the output to the model. Embedding dropout that sets an entire row of the word embedding matrix to **0** was also tuned, eliminating certain words in the embedding matrix. We also regularized hidden-to-hidden weight matrix using DropConnect [60]. At last, activation regularization (AR) and temporal activation regularization (TAR) coefficients were tuned. For all RS, BO, STNs, and $\Delta$-STNs, the search spaces for AR and TAR were $[0, 4]$ and we initialized them to 0.05. Similarly, all dropout rates were initialized to 0.05, and the search space was $[0, 0.95]$ for STNs and $\Delta$-STNs while it was $[0, 0.75]$ for RS and BO. We present the hyperparameter schedules found by $\Delta$-STNs in figure 13.

# I  Additional Results

We present additional experiments in this section.

## I.1  Hyperparameter Schedules

Because the hyperparameters are tuned online, STNs do not use a fixed set of hyperparameters throughout the training. Instead, it finds hyperparameter schedules that outperforms fixed hyperparameters [38]. We trained a multilayer perceptron on MNIST dataset and tuned the input dropout matrix with using STNs and $\Delta$-STNs. The same experimental configurations to those of

| Method | Valid | Test |
|---|---|---|
| $p = 0.5$, Fixed | 0.059 | **0.054** |
| $p = 0.5$ w/ Gaussian Noise | 0.057 | 0.059 |
| $p = 0.43$ (Final $\Delta$-STN Value) | 0.059 | 0.060 |
| STN | 0.058 | 0.058 |
| $\Delta$-STN | **0.053** | **0.054** |

**Table 4:** A comparison of validation and test losses on MLP trained with fixed and perturbed input dropouts, and MLP trained with STNs and $\Delta$-STNs.

(a) SimpleCNN

(b) AlexNet

**Figure 15:** The effect of using different perturbation scale on **(a)** SimpleCNN and **(b)** AlexNet for STNs and $\Delta$-STNs. $\Delta$-STN is more robust to a wider range of perturbation scale.

**Figure 16:** The effect of using different training and validation update intervals on $\Delta$-STNs

MNIST experiment (appendix H.2.1) were used, except that we only tuned a single hyperparameter and fixed the perturbation scale to 1. We compared the hyperparameter schedules found by STNs and $\Delta$-STNs with different initializations. As shown in figure 14, $\Delta$-STNs found the hyperparameter schedule more robustly compared to STNs. We further compared best validation loss and corresponding test loss achieved by $\Delta$-STNs and STNs in table 4. The fixed and perturbed dropout rates found by grid search and given by final $\Delta$-STN value were also compared. Our $\Delta$-STN was able to find the hyperparameter schedules more accurately and robustly as well.

## I.2    Sensitivity Studies

We show the sensitivity of $\Delta$-STNs to meta-parameters. Specifically, we investigated the effect of using different training and validation update intervals (figure 16), and different fixed perturbation perturbation scale (figure 15) on SimpleCNN and AlexNet. $\Delta$-STNs showed more robustness to different perturbation scale.