[Reviews · NeurIPS 2020]

Review 1

Summary and Contributions: The paper proposes a novel architecture and a training objective for “Self-Tuning Networks” (STN), which is hyper-network constructed for each layer of a neural network and is used to obtain an approximate solution to the inner level objective of a bi-level optimization problem. Hyper-parameter optimization is a typical example of a problem that takes a bi-level form, where the inner level searches for the optimal weights of the network while the outer level solves for the hyper-parameters. In contrast to other approaches, STNs offer the use of a gradient based method for optimization and do not require re-training of the models from scratch. The paper develops on top of STNs. It identifies the core inefficiencies within the method and proposes alternatives to mitigate such inefficiencies. Specifically, the paper proposes three changes. First, a centered parameterization for the best-response function which improves the condition number of the training loss function. By computing the Gauss-Newton Hessian, the paper shows that it indeed improves the condition number. Second, train the bias parameter $w_0$ of the STN with gradient descent on a regularized training loss and third, linearize the network which helps remove the non-linear dependence of the outputs on the weights of the network. Experiments on different datasets and networks provide evidence that the proposed approach, $\Delta-$STNs, converge faster, provide better accuracies and are more stable.

Strengths: The ideas are backed with sufficient mathematical analysis. The paper computes the approximate Gauss-Newton Hessian (Equations 4.4) and shows that a centered parameter schemes lowers the condition number (Equation 4.5), which is convincing. The paper provides a neat analysis in Appendix E on how the older update rule wouldn’t let $w_0$ converge to its correct solution. These mathematical demonstrations on simple objectives (such as a quadratic problem) provide better clarity of the method to the reader. The experiments presented in the paper validate the claims. A good list of experiments ranging from three layered neural networks to deep networks for image classification to recurrent neural network are presented. Moreover, an important set of hyper-parameters have been used such as variational dropout, drop-connect and activation parameters. It would have been interesting to also include a batch-normalization layers in their network and optimize its hyper-parameters, as batch-normalization layer is a standard layer in many modern deep networks. $\Delta-$STN’s, when compared to STN’s, converge faster, achieve higher accuracy and improve training stability. They seem to have potential to replace STNs for practical purposes.

Weaknesses: The whole idea of learning a parametric approximation of the best response function is not new. The key steps of the algorithm, for example to approximate the best-response function around the current hyper-parameter value and the joint optimization of hypernetwork and hyperparameters, are similar to that of STNs. The paper doesn’t explore a new research direction but appears to be an addition to the existing work on STNs. This is a weakness of the paper. The discussion on the interactions between the inner and outer optimization problems and their effect on uncentered parameterizations (lines 152 to 165) is not clear. In line 161, the paper claims that the gradients are well aligned early in training, why is it true? Does the inner-product becomes negative later in the training? Please elaborate on this discussion. I disagree with the claim made on line 249 that section 5.2 evaluates the scalability of their method. CIFAR10 and PTB are small scale problems! I’m not asking for newer experiments but at the very least to avoid the use of the word “scalability” in that section. It would be helpful to provide a comment or discussion on the performance on an ImageNet scale datasets. Similarly, a reader would be interested in knowing the usability of the method outside the domain of hyper-parameter optimization like neural architecture search (NAS). I believe an experiment for NAS would be out of the scope of the paper but a discussion on whether the method is applicable on other domains that also rely on the bi-level optimization formulation would be helpful. The toy experiments give the same message as the image classification experiments, i.e., they talk about efficiency, stability, etc. It would have been better to use the toy experiments for a different purpose, probably to isolate and observe the effects of centered parameterization alone or the effect of modified training objective alone. Currently, a baseline of such kind ‘centered-STN’ (which combines centered parameterization and modified objective) has been studied and it was helpful. The base networks used in the paper, VGG and AlexNet are not contemporary architectures. It would have been better to see maybe a ResNet18 architecture.

Correctness: The proofs provided in the supplement seem to be correct.

Clarity: The paper is very well written! The use of equation number of the format “Section Number <dot> Equation Number” helped to easily navigate through the paper. The proofs were easy to read. The idea to break Equation B.16 (supplement) and to evaluate each term individually made reading that section more convenient.

Relation to Prior Work: The paper has cited important prior works and discusses them in its related works section. In lines 63-65, the paper discusses two major approaches that also use gradient based optimization schemes. Please provide the pros and cons of these method in comparison to STNs. Section 3.3 (which discusses the best-response hyper-network) can be expanded. Specifically, it would be helpful to discuss why the standard STNs choose the objective in Equation 3.5 to optimize and why the best-response function was parameterized by STNs as shown in Equation 3.7. The supplement provides the training algorithm for standard STNs. Please indicate this in Section 3.3.

Reproducibility: Yes

Additional Feedback: ################### POST-REBUTTAL ######################### I thank the authors for their response. I find the plot that demonstrates the riddance of the spike based only off the centering modification very convincing and would suggest to include it in the main paper. Despite only a minor improvement in validation accuracy, it is convincing that the method benefits from stability and faster convergence. My concerns were addressed in the response and I have increased my score. ########################################################### Can the faster convergence and stability be obtained by using a good learning rate schedule ($\alpha_1$, $\alpha_2$, $\alpha_3$), thereby replacing the necessity of using the proposed $Delta-$STN? Why does the code for STN run faster than $\Delta-$STN per epoch? Will this have any effect on the wall-clock convergence time specifically for the plots in Figure 2? How good is the approximation for Guass-Newton Hessian in Equation 4.4? Under which circumstances would it be “wrong” to rely on this approximation? How significant is the effect of $G_w$ on the condition number of the Hessian in Equation 4.4? I anticipate that centering the hyper-parameters $\lambda$ would cause a much slower update on the $\lambda$. This is because the gradient on the validation loss would be smaller resulting in almost no change in $\lambda$ values. In this case, the algorithm would have to heavily depend on a good initialization of $\lambda$. The experiments don’t seem to indicate this behavior though, why? Do the experiments rely on a good initialization for $\lambda$? What is the primary advantage of using a linearized network in terms of the empirical results? Is it faster convergence alone or does it also help with the stability? Please mention the phrases “general” and “response” in the text of Section 4.4 and explain them. These phrases are used without any description in Equation 4.12. Please provide a comment on the standard deviation of the $lambda$ parameter in Figure 2. Does the standard deviation improve compared to STNs? Why is there a huge spike in the $\lambda$ parameter for STN in Figure 2? Can the spike be mitigated by reducing the learning rate parameter $\alpha_3$? Why are the loss values for $Delta-$STN and STN not plotted beyond 40000sec in Figure 5a? In Table 1, how much does the improvement in the loss value translate in the improvement in accuracy? Please mention this in the text of the paper as well. Can you elaborate on the term “warm-up” in the line 531 of the supplement? Do you maintain a fixed value for $\lambda$ in this phase? If so, I would expect the plots for STN and $\Delta-$STN to be identical during this phase (i.e., for the first x number of iterations) which does not seem to be the case. Line 590 in the supplement, Why do you use a different values for the penalty parameter $\tau$ in $\Delta-$STN and STN? For fair comparison, wouldn’t you want to set the same $\tau$ for both the methods? Some very minor errors: Line 59 “paralleize” -> “parallelize” Line 176 “the the” -> “the” Line 268 “CIFAR-10” -> “CIFAR-10.” <with a dot>


Review 2

Summary and Contributions: This paper identifies certain issues in training of STNs (Self-Tuning Networks) and propose a new algorithm called delta-STN by: 1) improving the conditioning of Gauss-Newton Hessian by reparametrizing the hypernetwork, 2) use of a modified training scheme for better updates with low bias and variance, and 3) the key idea of accurately estimating the best-response Jacobian instead of the full best-response function, which is made tractable by linearizing the network around the current parameters. The proposed delta-STN algorithm is empirically demonstrated to consistently outperform STNs.

Strengths: - Clear demonstration of convergence to optimal hyperparameter value in a toy linear regression problem. - Empirical evaluation against STN with convolutional networks in image classification benchmarks (MNIST, FashionMNIST and CIFAR10) and recurrent networks in language modelling (PBT) shows that the proposed algorithm (delta-STN) consistently achieves faster convergence and better performance to outperform STN. These results are promising even though they are on small scale problems. - The proposed method, like STNs, can be easily used as a drop-in replacement for existing networks.

Weaknesses: - Not able to tune the learning rate, which is an important hyperparameter. - The meta-parameters still have to be chosen manually. However, the ablation studies in Section G.2 show that the proposed method is not very sensitive to the update intervals. The sensitivity to entropy weight is not presented.

Correctness: Yes. The empirical methodology is same as STNs.

Clarity: They paper is well written and easy to read.

Relation to Prior Work: Yes, the relation to prior works and the contributions of this work are clearly discussed.

Reproducibility: Yes

Additional Feedback: Update after author response: I acknowledge that I have read the rebuttal.


Review 3

Summary and Contributions: This paper studies optimization dynamics and stability issues involving in training a hypernetwork to approximate the inner-loop optimization in a bilevel optimization problem, where the approach of bilevel optimization is used for learning optimal hyperparameters of a neural network. The hypernetwork particularly focused by this paper is a linear network of STN [39]. To improve training stability and convergence of STN, the paper proposes reparameterization of the hypernetwork with the trick of parameter centering; the paper justifies the reparameterization with its benefits for training the hypernetwork itself, and also for bilevel optimization. The paper also uses first-order Taylor approximation to linearize the precision at the current prediction of the hypernetwork. The paper conducts a few toy experiments to verify the efficacy of the proposed schemes for improving learning of STN.

Strengths: The paper presents Observations 1 and 2 to justify the proposed parameter centering for better training hypernetwork. Theorem 3 guarantees that for quadratic objective, the proposed reparameterization converges to the best-response Jacobian.

Weaknesses: It is good to justify the proposed scheme of parameter centering. However, the justification for its first benefit seems a common practice in analyzing training convergence of neural network, by analyzing its condition number of Hessian. The second justification is more interesting, since it is directly relevant to bilevel optimization, however, the second justification seems heuristic with no further theoretical analysis. Math notations of the paper can be improved. The current ones used in section 4.1 are kind of confusing readers. While experiments in sections 5.1 and 5.2 verify that the proposed method is better than STN, however, the improvements are rather limited. Furthermore, for results reported in Table 1, whether the reduced loss values translate as improved accuracies? The paper does not explain its broader impact at the end. I also have concerns on the impact of the contributions made in this paper.

Correctness: Seems correct. Due to less clear notations, some equations are not fully checked.

Clarity: The paper writing can be improved, especially for the math notations.

Relation to Prior Work: Yes, the paper builds upon previous hypernetwork of STNs. The new contributions concerning with better training of STNs are clearly stated.

Reproducibility: Yes

Additional Feedback: It seems that equation 3.1, and the very brief introduction of bilevel optimization of section 3.1 are not quite necessary. The relevant knowledge can be merged with the introduction of equation 3.2 for the specific bilevel optimization problem considered in this paper. When printing the paper, Figure 1 is not shown.


Review 4

Summary and Contributions: The authors propose three changes over self tuning networks: a) approximating the Jacobian of the best response function using a first-order Taylor expansion. b) using a centered parameterization for the STN hyper network. c) training the current weights of the network without sampling hyper parameters. The authors motivate their changes theoretically and empirically show significant improvements in performance when compared to both STN and random search and bayesian optimization.

Strengths: 1. The proposed changes are simple to implement and theoretically motivated. 2. Comparing with centered-STN provides evidence that approximating the Jacobian using a Taylor expansion is an integral component of the proposed method. 3. Thorough experiments show clearly that centered-STN improves over STN and Δ-STN improves over both. 4. The paper is well written and easy to follow.

Weaknesses: 1. Random search and Bayesian optimization can also optimize hyper parameters related to the training procedure. How do STNs compare?

Correctness: The claims in the paper as well as the empirical methodology are correct.

Clarity: The paper is clearly written.

Relation to Prior Work: The related work section is thorough enough. A discussion regarding the differences between the capabilities of STNs and some traditional hyper parameter optimization methods such as random search could be added.

Reproducibility: Yes

Additional Feedback: One thing I am missing from the paper is a discussion about the limitations of STNs in general and possibly a comparison with hyper parameters that cannot be optimized with STNs. For instance, what is the score achieved when training a ResNet with Δ-STN but with a fixed learning rate schedule and one optimized with random search? There are also the following typos: 1. Line 105: alternative should be alternating 2. Line 160: λ should be with a bar on top Post Rebuttal ------------------- I thank the authors for their rebuttal. After reading it and discussing with fellow reviewers I am keeping my initial score.

[Author Response · NeurIPS 2020]

We thank the reviewers for their detailed reviews and constructive feedbacks. We are encouraged that the reviewers
found our paper well-written & easy to follow [**R3 R4 R6**], theoretically motivated [**R3 R6**], empirically convincing
[**R3 R4 R6**], and novel [**R3**]. The major concerns and questions are addressed below; we will incorporate all suggestions
in the next revision and improve the notation (also include a table of notation) as suggested by **R5**.

[**R3 R5**] **Undesirable bilevel dynamics for uncentered parame-**
**terization.** In support of our claim that the training and validation
gradients are well aligned early in training, the left-hand figure shows
(for linear regression and an MLP) that the angle between the training
and validation gradients starts close to 0 and increases throughout
training. Based on the analysis in Section 3.1 (line 152), this align-

ment causes "spikes" in the hyperparameter values (Fig. 2 on page 7 and Fig. 6 on page 17), even with smaller learning
rates. The right-hand figure decouples the centered parameterization from all other changes to the training objective;
the centering by itself eliminates the spike, as our analysis predicts. This effect and the poor conditioning of the
Gauss-Newton Matrix, limit the optimization performance of uncentered representations.

[**R3**] **Smaller validation gradient $\Rightarrow$ sensitive to initial hyperparameters?** The uncentered parameterization indeed
results in larger hypernetwork gradients, but in a way that is unhelpful for the optimization as described above. Hence,
centered parameterizations are *less* sensitive to hyperparameter initializations.

[**R3 R6**] **Applicability to other bilevel problems and limitations of the STN approach.** STNs and $\Delta$-STNs consti-
tute a third general approach to gradient-based bilevel optimization, in addition to implicit differentiation (ID) and
unrolling. In relation to those two approaches, ($\Delta$-)STNs are much more efficient, because they amortize the inner
optimization and the response Jacobian. The range of applicability is slightly more restricted, as STNs require a single
(rather than per-example) inner objective and (like ID but unlike unrolling) require that the outer variables parameterize
the training objective. We note that NAS falls into this category and STN approaches can be applied. We will add this
discussion to the paper.

[**R5**] **Societal impact.** As almost any application of deep learning involves regularization hyperparameters, and
hyperparameter tuning is one stage of a much longer pipeline, any discussion of societal impacts would necessarily be
speculative. One predictable impact is to lessen the need for massive computing resources to tune hyperparameters.

[**R3 R4 R6**] **Learning rates and other hyperparameters.** Out of the three aforementioned gradient-based approaches
to bilevel optimization — implicit differentiation, unrolling, and ($\Delta$-)STNs — only unrolling is applicable to learning
rate tuning. However, even for unrolling, short-horizon objectives suffer from a severe bias [Wu et al, 2018] which
makes unrolling impractical for learning rate adaptation unless one uses huge amounts of computation (e.g. [Metz
et al., 2019]). Learning rate adaptation remains a hard problem in general. We note that it would be natural to wrap
Bayesian optimization (or random search) around the $\Delta$-STN to optimize the remaining hyperparameters that $\Delta$-STNs
are inapplicable to. BO suffers from high-dimensional search spaces and pushing the responsibility for regularization
hyperparameters onto the $\Delta$-STN can significantly reduce the dimensionality of the BO search space.

[**R3 R4**] **Impact of meta-parameters.** While it is always difficult to rigorously quantify
the ease of tuning, we found $\Delta$-STNs to perform well with a default set of meta-parameters
(e.g. fixed perturbation scale, choices of $\alpha_i$) across all tasks we investigated, unlike STNs.
$\Delta$-STNs are more forgiving of large perturbation scales than STNs, because they do not need
to model the entire response function over the likely range of hyperparameters. Furthermore,
$\Delta$-STNs are less sensitive than STNs to initial hyperparameters (e.g. figure 10), for the

reasons described in Section 3.1. The right-hand figure shows the effect of different entropy weights, where we grid
searched over $\{10^{-2}, 10^{-3}, 10^{-4}\}$ on all experiments. We will provide a more thorough sensitivity analysis in the next
revision.

[**R3 R5**] **Validation accuracy, and significance.** $\Delta$-STNs improve over STNs on validation accuracy as well as loss:
on AlexNet and VGG16, they improve accuracy by $0.5\%$ and $1.3\%$, respectively. In Table 1, we showed that $\Delta$-STNs
consistently outperform STNs with $4\% \sim 13\%$ improvements in the validation objective across all tasks. We believe
that these improvements are significant, since we are using the same set of regularization and data augmentation
hyperparameters, without modifying the architecture or introducing new techniques. We also showed that $\Delta$-STNs are
more stable and converge faster compared to STNs.

[**R3**] **Scalability.** We modified the statement made on line 249. On a ResNet-32 ImageNet classifier, the $\Delta$-STN
requires only a factor of 1.8 more time per epoch compared to training a single network (and this overhead can be made
arbitrarily small by updating the hyperparameters less frequently). We did not include ImageNet experiments because,
for proper baseline comparisons, it would cost thousands of dollars, but there is no computational obstacle to running
$\Delta$-STNs on ImageNet.

[Meta-Review · NeurIPS 2020]

Originally, the paper scores were: 7,6,4,8, all with relatively high confidences. The most negative one, Reviewer #5, concerned about some of the justifications, limited experimental improvements, and potential impacts. During the discussion, Reviewer #5 acknowledged that his/her concerns were all addressed except the impact issue, and increased his/her score to 5. Reviewer #3 echoed and increased his/her score too. Reviewer #6 confirmed the contribution of the paper. So the AC recommended acceptance. The AC also had a quick reading on the paper and had the follow comments: Pros: By introducing the centered parameterization, two-state hypernetwork estimation and lower-level objective linearization techniques, this work does provide extensions and improve the performance of STN. A series of experiments have also demonstrated the performance of these extensions. Cons: First, authors only provided guarantees (Theorem 3) for the centered parameterization and two-state updating extensions. While no results are proved for the linearization trick, which actually plays more important role in speeding up the STN computation. Second, since a series of approximations have been introduced to the STN process, it is unclear why the proposed \Delta-STN can still achieve higher accuracy than standard STN in these experiments. The AC believed that detailed ablation results are necessary in the experimental part. Minor suggestions: The structure and organization should be improved. For example, Sec 2 is too brief, while most materials in Sec 3 are quite standard and should be refined. Fig 1 is also redundant.